# Discovery and substrate specificity engineering of nucleotide halogenases

Jie Ni[1], Jingyuan Zhuang[1], Yiming Shi[1], Ying-Chih Chiang [2] &
Gui-Juan Cheng [1] ✉

C2′-halogenation has been recognized as an essential modification to enhance the drug-like properties of nucleotide analogs. The direct C2′-halogenation of the nucleotide 2′-deoxyadenosine-5′-monophosphate (dAMP) has recently been achieved using the Fe(II)/α-ketoglutarate-dependent nucleotide halogenase AdaV. However, the limited substrate scope of this enzyme hampers its broader applications. In this study, we report two halogenases capable of halogenating 2′-deoxyguanosine monophosphate (dGMP), thereby expanding the family of nucleotide halogenases. Computational studies reveal that nucleotide specificity is regulated by the binding pose of the phosphate group. Based on these findings, we successfully engineered the substrate specificity of these halogenases by mutating second-sphere residues. This work expands the toolbox of nucleotide halogenases and provides insights into the regulation mechanism of nucleotide specificity.

Late-stage functionalization of C−H bonds has proven to be an efficient method for modifying complex molecules[1–5]. Among the various reactions in this field, late-stage C−H halogenation is particularly valuable as it allows for the manipulation of bioactivity and the installation of reactive handles for subsequent transformations[6,7]. Despite intense efforts, the halogenation of unactivated $sp^3$ C−H bonds remains a challenging task due to the difficulties in finding a catalyst that enables high reactivity and selectivity under mild reaction conditions. In this regard, the non-heme Fe(II)/α-ketoglutarate-dependent halogenases (αKGHs) have emerged as competent biocatalysts for this difficult task[8,9]. αKGHs employ a high-valent iron-oxo intermediate to selectively activate $sp^3$ C−H bonds, generating a substrate radical that subsequently reacts with a halide ligand to yield the halogenated product[10,11]. A series of αKGHs acting on carrier-protein-tethered substrates (SyrB2 family)[12–17], alkaloid natural products (WelO5 family and DAH)[18–24], amino acids (BesD family)[25,26], and nucleotides (AdaV family)[27,28] have been discovered and attracted extensive attention.

Nucleotide and nucleoside analogs (NAs) are valuable tools and therapeutics that interfere with vital biological processes, such as DNA replication. They are widely used in the treatment of viral and oncological diseases (Fig. 1a)[29–32]. Modification at the C2′ position of the deoxyribose is an effective strategy to enhance the potency and pharmacodynamic properties of NAs[33,34]. Chemical C2′-modification methods often require multiple protecting group manipulations, resulting in lengthy synthetic routes (Fig. 1b)[35–38]. The recent discovery of the Fe(II)/α-ketoglutarate-dependent radical nucleotide halogenases, AdaV, provides a late-stage $sp^3$ C−H halogenation method that enables direct modification of the deoxyribose ring at the C2′ position with high regio- and stereoselectivity[27,39] (Fig. 1c). The introduced halogen atom could serve as a reactive handle for subsequent modifications through substitution, holding great promise for advancing the field of nucleotide modification. However, AdaV preferentially accepts dAMP as a substrate[27], limiting its applications as a nucleotide halogenation tool. To address this limitation, new αKGHs that can act on alternative nucleotide substrates are highly sought after.

In this work, we reported the discovery and characterization of two Fe(II)/α-ketoglutarate-dependent halogenases, which are capable of halogenating dGMP. Through computational and experimental studies, we elucidated the molecular basis for the nucleotide specificity of dGMP and dAMP halogenases. Based on the nucleotide discrimination mechanism, we successfully altered their substrate specificity.

[1]Warshel Institute for Computational Biology, School of Medicine, The Chinese University of Hong Kong, Shenzhen 518172 Guangdong, China. [2]Kobilka Institute of Innovative Drug Discovery, School of Medicine, The Chinese University of Hong Kong, Shenzhen 518172 Guangdong, China. ✉e-mail: chengguijuan@cuhk.edu.cn

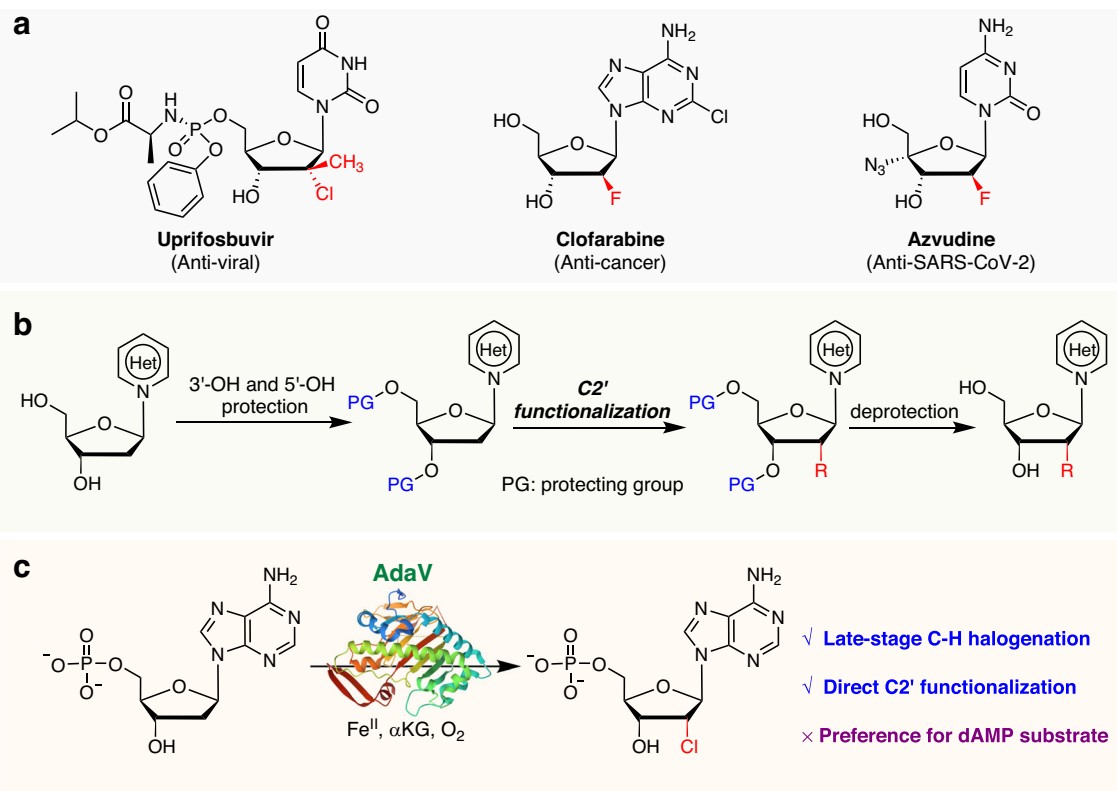

**Fig. 1 | Selected nucleotide and nucleoside analogs (NAs) and C2′-modification methods. a** Halogenated NAs for clinical use. **b** Chemical method for C2′-modification of nucleoside analogs. **c** AdaV enables the direct C2′-halogenation of dAMP.

## Results

### Identification of dGMP nucleotide halogenases

To search for potential nucleotide halogenases candidates, we performed a sequence-based BLAST (basic local alignment search tool) analysis in NCBI (national center for biotechnology information) using AdaV as a query. The obtained homologs (with a 30% identity cutoff) were filtered based on the characteristic HXG/A motif of the αKGHs. The generated ten candidates include three recently identified dAMP halogenases[39] (DI270, ADK38, and BJP25 in Fig. 2a and Supplementary Table 1), which were thus excluded. The remaining seven candidates were submitted for maximum-likelihood phylogenetic analysis with reported free-standing αKGHs[18–20,24,25,27,39,40]. As depicted in Fig. 2a, five candidates (labeled as 1, 3, 4, 6, and 7), which share over 70% sequence identity with AdaV, clustered within the same clade as the characterized dAMP halogenases. This clustering strongly suggests their classification as dAMP halogenases. Intriguingly, the last two candidates (9 and 10) formed a distinct clade in the constructed phylogenetic tree, indicative of their potential as nucleotide halogenases with differing substrate specificity. The two putative halogenases are designated VaNTH and CtNTH as they are from *Virgisporangium aurantiacum*[41] and *Catellatospora tritici*[42], respectively.

To validate the in silico analysis, we cloned, heterologously expressed and purified the two putative nucleotide halogenases, and tested their activity against a panel of natural nucleotides and nucleotide analogs (Supplementary Figs. 2, 3). HPLC and LC-MS analysis demonstrated that VaNTH and CtNTH are active toward dGMP, dIMP (2′-deoxyinosine monophosphate), and dAMP (Fig. 2b–d, Supplementary Figs. 4–7). But the two enzymes display a clear preference for dGMP as a substrate, differing from AdaV in nucleotide specificity. NMR experiments confirmed that the Cl atom adds to C2′ of dGMP from the same side of 3′-OH group (Supplementary Figs. 8–11). Analysis of the kinetic parameters revealed that VaNTH ($k_{cat}/K_m = 18.6$ mM$^{-1}$ min$^{-1}$) and CtNTH ($k_{cat}/K_m = 17.8$ mM$^{-1}$ min$^{-1}$) exhibited a >43-fold increased $k_{cat}/K_m$ when compared to AdaV ($k_{cat}/K_m = 0.44$ mM$^{-1}$ min$^{-1}$, Supplementary Fig. 12).

The introduction of various substituents at the C2′ positions of deoxyribose sugar has proven to be an effective strategy to manipulate the activity of functional nucleotide derivatives. We, therefore, tested the ability of VaNTH and CtNTH to install alternative anions to the C2′ position of dGMP using a range of halide and pseudohalide anions, including Br$^-$, I$^-$, N$_3^-$, NO$_2^-$, OCN$^-$, and SCN$^-$. It was found that I$^-$ and NO$_2^-$ were unreactive. Br$^-$ and N$_3^-$ were successfully incorporated to the substrate with inferior reactivity compared to Cl$^-$ (Supplementary Figs. 13, 14). These findings are consistent with the anion promiscuity reported for other αKGHs, such as SaDAH[24] and BesD[25]. In all these reactions, the hydroxylation of dGMP was observed as a background reaction, which led to a trace amount of GMP byproduct. It is intriguing that the addition of OCN$^-$ or SCN$^-$ significantly increased the efficiency of hydroxylation reaction (Supplementary Fig. 15). Due to the different steric and electronic effects of OCN$^-$ and SCN$^-$, the replacement of Cl$^-$ by these anions alters the property of the catalytic center, which may facilitate the hydroxylation reaction.

Overall, through bioinformatic and biochemical analysis, we have identified two dGMP chlorinases that also display bromination and azidation activities. Given the significant sequence identity (67.3%) and highly conserved active site residues between VaNTH and CtNTH (Supplementary Fig. 20), we chose CtNTH as the representative dGMP halogenase for further investigations.

### Binding mode and mechanism of nucleotide specificity

The newly identified enzyme CtNTH displays distinct nucleotide specificity with AdaV, despite their high sequence identity (51.0%, Supplementary Table 1). Understanding their substrate preference can provide valuable information for protein engineering to alter the nucleotide specificity. To this end, we first examined the substrate binding modes by molecular dynamic (MD) simulations of CtNTH and

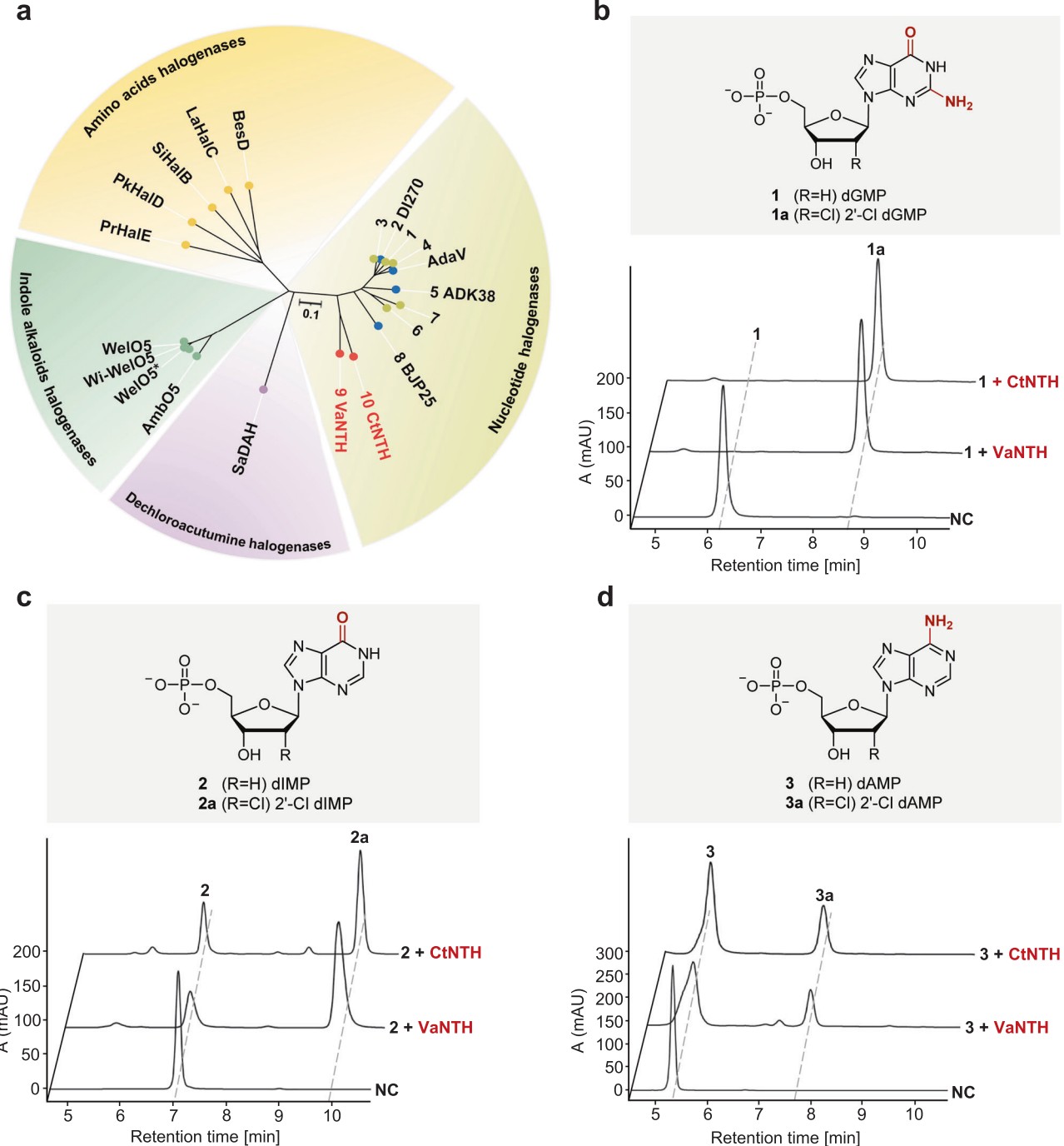

**Fig. 2 | Identification and characterization of nucleotide halogenases.**
**a** Phylogenetic analysis of reported Fe(II)/α-ketogluturate-dependent halogenases and putative halogenases. The (putative) nucleotide halogenase were assigned numerical labels based on their sequence identity ranking with AdaV. Blue dots: reported dAMP halogenases; Olive dots: putative dAMP halogenases; Red dots: putative nucleotide halogenases with different substrate specificity. **b, c, d** HPLC analysis of enzymatic reactions catalyzed by VaNTH and CtNTH with substrate **1** (dGMP), **2** (dIMP), and **3** (dAMP). The experiments were repeated independently three times with similar results. NC (negative control): standard reaction without enzyme.

AdaV enzymes with their preferred substrates. For each system, three replicates of 500 ns MD simulations with the same initial structure were performed. The initial structure of AdaV/dAMP used in MD simulations was obtained from a previously reported crystal structure[39]. For CtNTH/dGMP system, the 3D structure of CtNTH was built by homology modeling using AdaV as a template, and dGMP was docked into the active site of CtNTH. The top-ranked pose, which is similar to the binding pose in the crystal structure of AdaV/dAMP, was used as the initial structure for MD simulations (Supplementary

Fig. 21). During the MD simulations of both CtNTH/dGMP and AdaV/dAMP systems, we observed two interconvertible substrate binding modes (denoted as mode 1 and 2) that differ in the orientation of the phosphate group (Figs. 3, 4).

In the binding mode 1 of CtNTH/dGMP system, the phosphate group forms hydrogen bonds with Arg178 and Lys108 (Fig. 3a). The nucleobase moiety is fixed by the π-π interaction with His195 and the hydrogen bond with Asp106. The deoxyribose ring of dGMP is closely bound to Fe=O center via a stable hydrogen bond between 3′-OH and

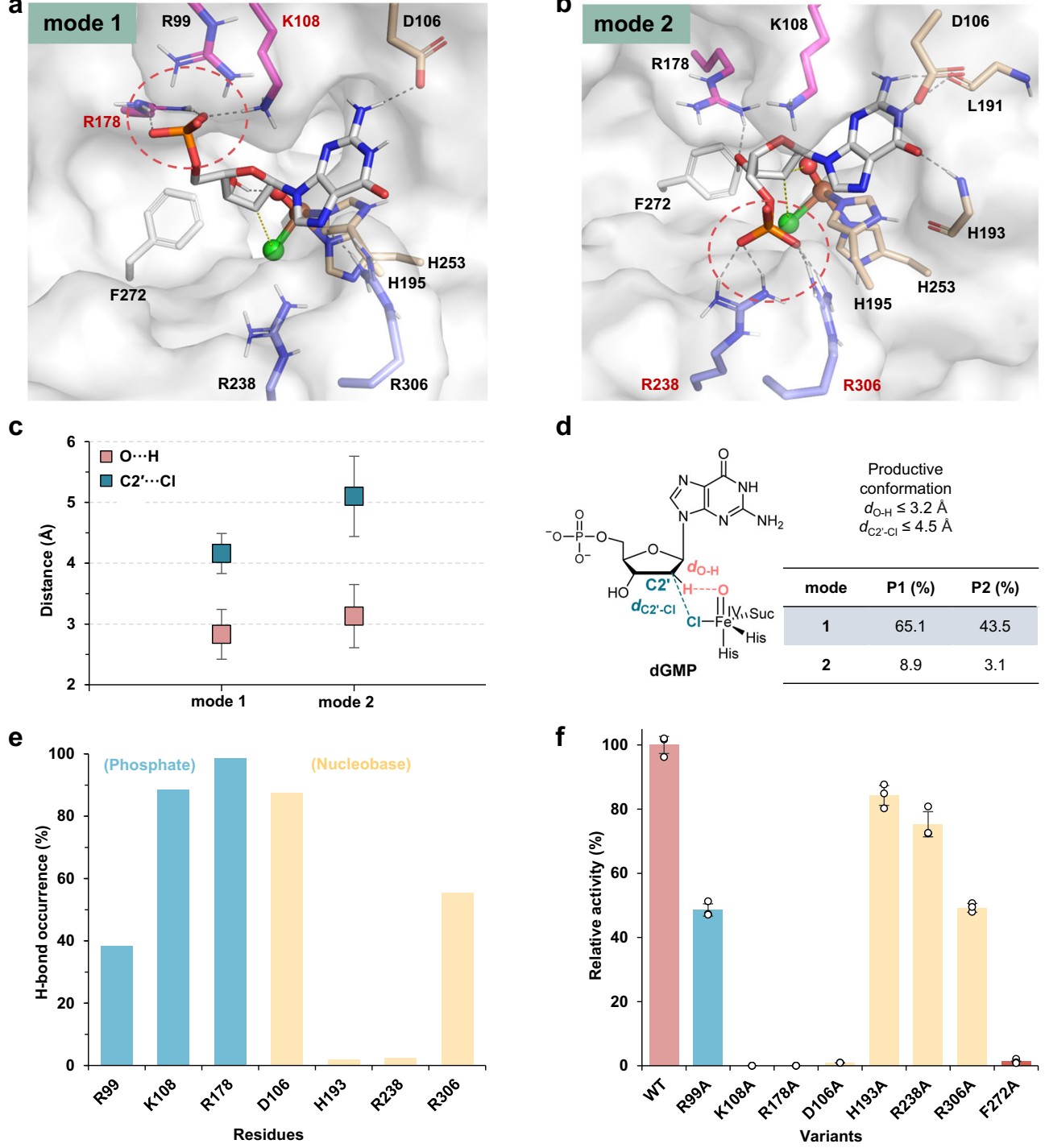

**Fig. 3 | Binding mode analysis of CtNTH/dGMP system by MD simulations and mutagenesis. a** Representative structure of binding mode 1. The iron atom (brown), oxo group (red), and chloride (green) are shown as spheres. Dotted cycles highlight the phosphate binding site. **b** Representative structure of binding mode 2. Arg99 is too far away to be included in the active site-focused figure. The figure including Arg99 can be found in SI (Supplementary Fig. 23c). **c** Statistical analysis of distances between reacting atoms, i.e., $d_{C2'-Cl}$ and $d_{O-H}$ (in Å). Data are presented as mean ± s.d. Square symbols represent the mean distances and the error bars indicate the standard deviation of the mean. **d** Statistical analysis of substrate binding modes and productive conformations. P1 represents the percentage of binding modes in total frames. P2 denotes the percentage of productive conformations for each mode in total frames. Suc stands for succinate. **e** Hydrogen bond occurrence between the substrate dGMP and the active site residues in mode 1. **f** Alanine mutation experiments of CtNTH. $n = 3$ biologically independent experiments. Data are presented as mean ± s.d. Source data are provided in Source Data file.

the oxo group. In addition, Phe272 positions the C2′ of deoxyribose ring to the proximity of the oxo and Cl groups[43]. Indeed, statistical analysis of essential distances of reacting atoms (Fig. 3c, Supplementary Table 2) suggests that the 2′-deoxyribose moiety of dGMP is

tightly bound to the catalytic iron center ($d_{O-H} = 2.8 \pm 0.4$ Å, $d_{C2'-Cl} = 4.2 \pm 0.3$ Å). According to a previous study[44], the conformations with $d_{O-H} \leq 3.2$ Å and $d_{C2'-Cl} \leq 4.5$ Å were considered as catalytic competent for the halogenation reaction and thus defined as productive

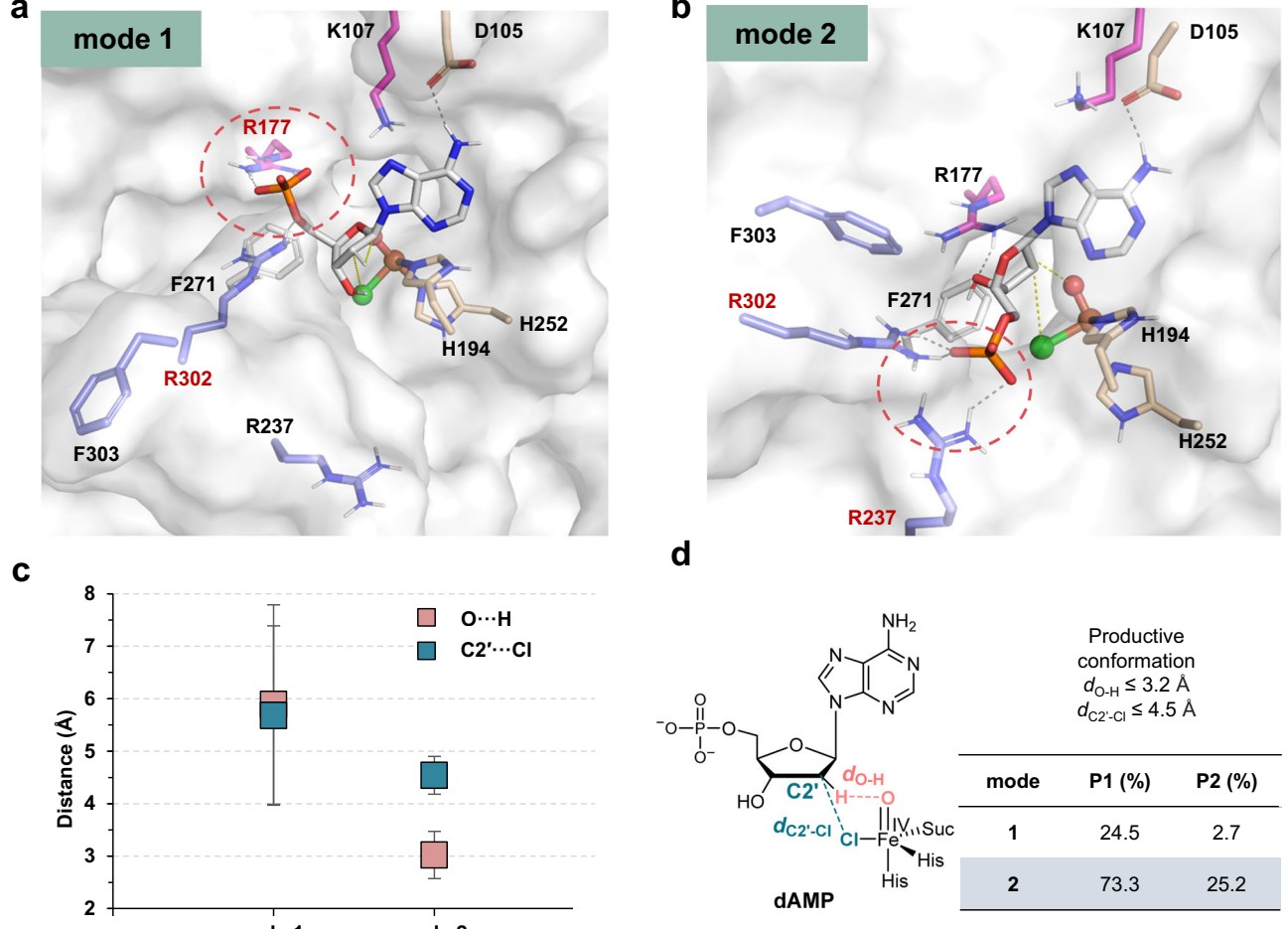

**Fig. 4 | Binding mode analysis of AdaV/dAMP system by MD simulations.**
**a** Representative structure of binding mode 1. The iron atom (brown), oxo group (red), and chloride (green) are shown as spheres. Dotted cycles highlight the phosphate binding site. **b** Representative structure of binding mode 2. **c** Statistical analysis of distances between reacting atoms, i.e., $d_{C2'-Cl}$ and $d_{O-H}$ (in Å). Data are presented as mean ± s.d. Square symbols represent the mean distances and the error bars indicate the standard deviation of the mean. **d** Statistical analysis of substrate binding modes and productive conformations. P1 represents the percentage of binding modes in total frames. P2 denotes the percentage of productive conformations for each mode in total frames. Source data are provided in Source Data file.

conformations. It was found that mode 1 is the dominant binding mode (65.1%) and its productive conformations accounts for 43.5% of the total frames. In contrast, in binding mode 2, the phosphate group flips downward and forms stable hydrogen bonding interactions with Arg238 and Arg306 (Fig. 3b, Supplementary Fig. 32). The significant movement of the phosphate group, which is attached to the deoxyribose moiety through a short methylene linker, alters the orientation of the deoxyribose moiety. Consequently, the hydrogen bond between 3'-OH and the oxo group is disrupted, and C2' is positioned at a considerable distance from Cl ($5.1 \pm 0.7$ Å). Statistical results show that mode 2 only occurs during 8.9% of the simulation time and its productive conformations only represent 3.1% of the total frames (Fig. 3d). Thus, these results collectively demonstrate that mode 1 is more likely to be the productive binding mode of dGMP for halogenation reaction.

Consistent with the computational analysis, the replacement of Lys108 and Arg178 with alanine completely abolished halogenation activity (Fig. 3e, f), confirming their crucial roles in substrate binding and supporting mode 1 as the productive binding mode. In contrast, the mutation of Arg238 to alanine retained 75% of the activity observed in the wild-type enzyme, supporting that mode 2 is not the major binding mode for dGMP. Moreover, the mutation results of other binding site residues are also in good agreement with mode 1. For instance, the alanine mutations of Asp106 and Phe272 which are

involved in the tight binding of substrate in mode 1 eliminated halogenation activity. Arg99 and Arg306 which are involved in the dynamic binding of dGMP in mode 1 (38.4% and 55.4% hydrogen bond occupancy, respectively) maintained medium activity after being mutated to alanine. His193, which barely interacts with the substrate in mode 1, exhibits high activity after being mutated to alanine. In summary, the computational analysis and mutation experiments together support that mode 1 is the productive binding mode of dGMP for halogenation reaction in CtNTH.

Similar binding modes were also observed in the MD simulations of the AdaV/dAMP system. However, statistical analysis and mutation experiments[39] suggest that the halogenation of dAMP prefers mode 2 as the productive binding mode. In mode 2, the deoxyribose ring of dAMP is positioned close to the oxo and Cl groups ($d_{O-H} = 3.0 \pm 0.5$ Å, $d_{C2'-Cl} = 4.5 \pm 0.4$ Å), allowing for H abstraction and chlorine transfer (Fig. 4b, c). By contrast, in mode 1, the H and C2' atoms are ~5.9 and 5.7 Å away from the oxo and Cl groups, respectively, which is not conducive to the halogenation reaction (Fig. 4a, c). Furthermore, unlike the CtNTH/dGMP system, mode 2 becomes the dominant binding mode (73.3%) in the AdaV/dAMP system. The productive conformations in mode 1 and 2 represent 2.7% and 25.2% of the total frames (Fig. 4d), respectively, providing supports for mode 2 as the productive binding mode. This finding is further supported by a

previous alanine mutation experiment of Arg237[39], which resulted in a significant decrease of chlorination activity.

The divergent productive binding modes observed in AdaV/dAMP and CtNTH/dGMP systems can be explained by the fact that the amino groups of the adenine and guanidine moieties are located at different positions on the purine ring, C6 and C2 respectively. The interaction between the amino group and the aspartate residue influences the orientations of guanine and adenine, leading to the placement of the deoxyribose moiety of dGMP and dAMP at distant positions within the same binding mode. For instance, in comparison to dGMP in mode 1 (Fig. 3a), the guanine ring of dAMP flips over, causing C2′ of the deoxyribose moiety to move away from the catalytic iron center (Fig. 4a). As a result, dAMP adopts an unproductive conformation in mode 1 of AdaV/dAMP.

The above analysis of CtNTH and AdaV enzymes with their preferred substrates demonstrates that modes 1 and 2 are the dominant binding modes as well as the productive binding modes for CtNTH/dGMP and AdaV/dAMP, respectively. Then, we performed MD simulations of CtNTH and AdaV with their inferior substrates (Supplementary Figs. 26, 31) to understand their substrate preference. The analysis of the CtNTH/dAMP system suggests that mode 1 remains the dominant binding mode (74.4%, Supplementary Table 3). However, mode 1 is not conducive to the halogenation reaction of dAMP (P2 = 2.1%), which explains the low activity of CtNTH towards dAMP. Similarly, mode 2 remains the dominant binding mode in the AdaV/dGMP system (71.1%). However, its productive conformations only occur 2.9% of the simulation time, resulting in low activity towards dGMP. In summary, the comparative analysis of CtNTH and AdaV with dGMP and dAMP substrates reveals two crucial points. Firstly, the binding pose of the 5′-phosphate group plays a critical role in controlling the reactivity of the nucleotide substrate. This finding aligns with a previous study that highlighted that the 5′-phosphate moiety in dAMP is essential for enzymatic activity of AdaV[27]. Secondly, CtNTH and AdaV exhibit distinct preferences for substrate binding modes. CtNTH favors mode 1, which facilitates the halogenation of dGMP, whereas AdaV favors mode 2, which facilitates the reaction of dAMP. Therefore, the substrate specificity is regulated by the binding modes, i.e., the binding poses of the phosphate group.

## Engineering of nucleotide specificity

Nucleobase specificity engineering offers a direct way to expand the substrate scope of nucleotide-binding enzymes[45,46]. However, it remains a challenging task due to the complex interaction network between the nucleotide and protein[47]. Specifically, controlling the selectivity between adenine (A) and guanine (G) is particularly difficult because of their similar structures[48,49]. On the other hand, the nucleotide-enzyme interaction is commonly regarded as the structural basis for the substrate specificity. Therefore, the engineering typically focuses on the nucleotide-binding residues, particularly those interacting with the base moiety, as nucleotides differ in their base structures. However, our simulations show that the nucleotide specificity of CtNTH and AdaV is regulated by the binding pose of the phosphate group. This insight led us to explore the potential for engineering nucleotide specificity by mutating residues located between the two phosphate-binding sites, which could potentially influence the binding modes distribution.

To this end, we aligned the sequences of dGMP halogenases with the reported dAMP halogenases to compare the residues located between the two phosphate-binding sites (Fig. 5a). As highlighted in Fig. 5b, these residues are located at the C-terminal α-helix, three antiparallel β-sheets (β5, β6, and β11), and their connecting loops. Among these residues, we specifically focused on the sites that distinguish between dGMP and dAMP halogenases while remaining conserved within each enzyme subfamily. Following this criterion, we identified six residues in CtNTH: Ile176, Val273, His274, Val301, Gly304, and Leu305, which correspond to Asn175, Ala272, Tyr273, Ile299, Arg302, and Phe303, respectively, in AdaV (Fig. 5b).

To pinpoint the specific residues involved in substrate selectivity, we constructed six single mutants (I176N, V273A, H274Y, V301I, G304R, and L305F) for CtNTH. These mutations were designed to replace the original residues in CtNTH with the corresponding amino acids found in the dAMP halogenases. The endpoint analysis suggests that I176N, V273A, V301I, G304R, and L305F do not affect the substrate specificity as they still exhibit clear preference towards dGMP (Supplementary Fig. 33). Intriguingly, H274Y alters the substrate preference of CtNTH, showing increased activity towards dAMP compared to dGMP (Fig. 5c). This suggests that His274 plays a role in substrate selectivity. The full steady-state analysis of CtNTH$^{H274Y}$ unveils that the altered substrate preference is primarily due to the substantial decrease in activity towards dGMP. Compared to wild-type CtNTH, the $k_{cat}$ decreases by 5-fold, and $K_m$ increases by 8-fold, resulting in a substantial decrease (38-fold) in catalytic efficiency ($k_{cat}/K_m$) towards dGMP for the H274Y variant. By comparison, CtNTH$^{H274Y}$ exhibits a 2-fold increase in catalytic efficiency towards dAMP. Though the activity-enhancement effect is less significant, the catalytic efficiency of CtNTH$^{H274Y}$ towards dAMP is already higher than the natural dAMP halogenase AdaV by 4-fold. Efforts to further enhance the activity of CtNTH towards dAMP by mutating His274 to other amino acids (Phe and Asn) or adding a second mutation to H274Y (I176N_H274Y, V273A_H274Y, and H274Y_V301I) were not successful (Supplementary Figs. 34, 35).

We conducted similar single mutation experiments to introduce the corresponding amino acids found in dGMP halogenases into AdaV. It was found that AdaV is highly sensitive to mutation as the designed mutants (N175I, A272V, Y273H, I299V, R302G, and F303L) significantly diminish halogenation activity towards dAMP (Supplementary Fig. 37). Double mutants, R302G_F303L and A272V_Y273H, also eliminate the enzymatic activity (Supplementary Fig. 38). All these mutations abolish the activity towards dGMP, except for F303L, which yields trace halogenation product. This prompted us to investigate other aliphatic amino acids at this site, including F303I, F303V, and F303A (Supplementary Fig. 39). To our satisfaction, the F303V variant significantly increases the activity towards dGMP and alters the substrate preference, as depicted in Fig. 5c. The remaining two variants exhibit a slight increase in activity towards dGMP. Interestingly, the full steady-state analysis reveals that the F303V mutation enhances the catalytic efficiency of AdaV towards its natural substrate by 5-fold, which differs from the effect observed in CtNTH$^{H274Y}$. For wild-type AdaV, the halogenation activity towards dGMP is too low to measure its kinetic parameters. However, the F303V mutation significantly improves the catalytic efficiency of dGMP ($k_{cat}/K_m = 4.0 \pm 0.1$ mM$^{-1}$min$^{-1}$), which is now ~2-fold of that for dAMP ($k_{cat}/K_m = 2.3 \pm 0.05$ mM$^{-1}$min$^{-1}$). Efforts to further enhance the activity of AdaV towards dGMP by combining an additional mutation, R302G, to the F303V variant were unsuccessful.

## Role of the second-sphere residues

To elucidate how second-sphere residues, H274 and F303, regulate the nucleotide specificity, we performed MD simulations of CtNTH$^{H274Y}$ and AdaV$^{F303V}$ with dGMP and dAMP substrates. Our MD simulation results indicate that the H274Y mutation induces a shift in the substrate binding mode of CtNTH from mode 1 to mode 2 (Fig. 6a). This alteration in binding mode is likely because the larger tyrosine residue positions Phe272 closer to Arg178, thereby disfavoring mode 1 (Fig. 6c, d).

As listed in Fig. 6a, the H274Y mutation increases the ratio of mode 2 from 20.4% to 66.6% in the CtNTH/dAMP system, accounting for its increased activity towards dAMP. Mode 2 is supported by the R238A mutation experiment of CtNTH$^{H274Y}$, where only 4.9% of the chlorination activity was retained for dAMP (Supplementary Fig. 45).

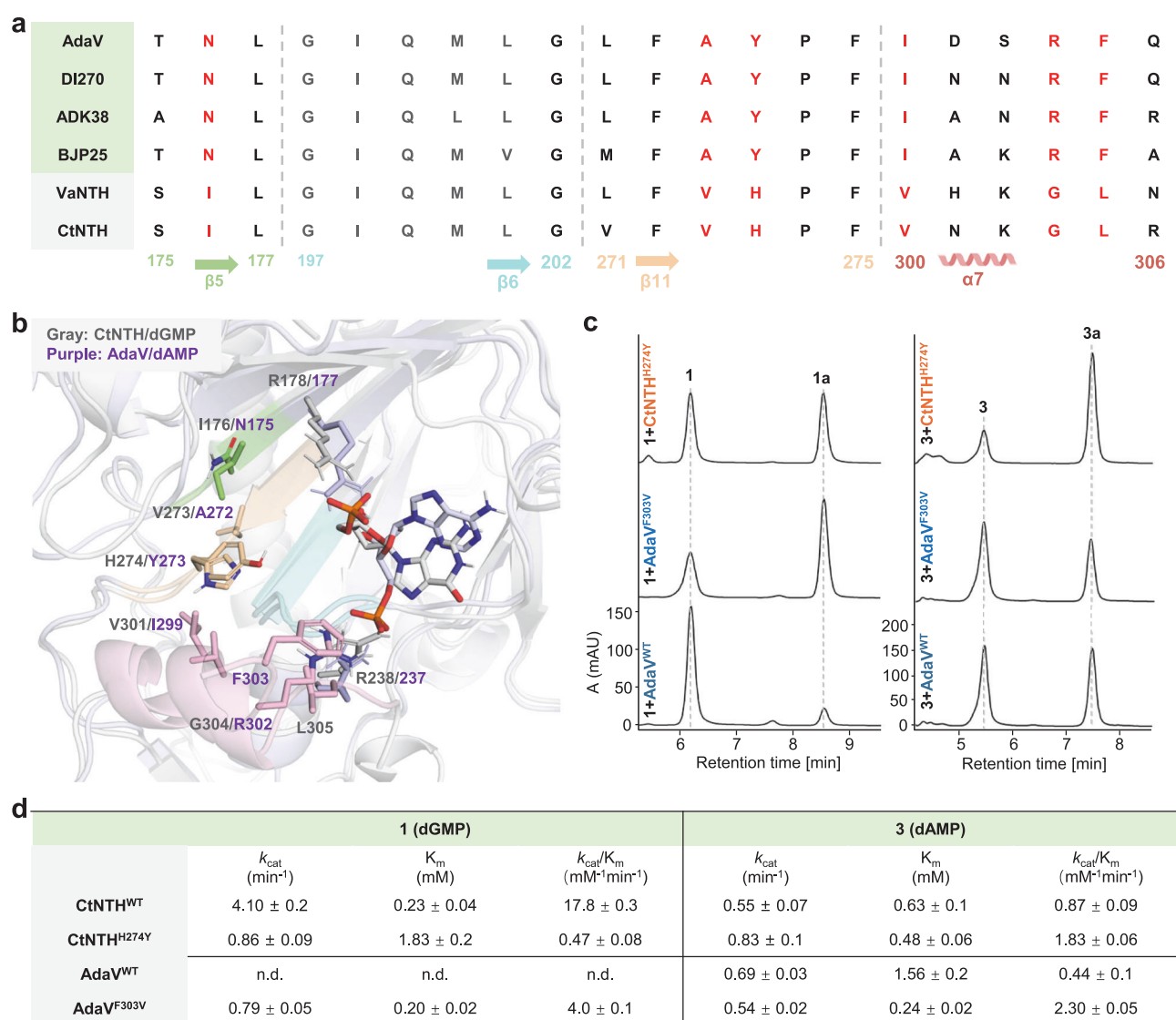

**Fig. 5 | Nucleotide specificity engineering of CtNTH and AdaV. a** Alignment of residues located between the two phosphate-binding sites in dGMP halogenases and reported dAMP halogenases. **b** Structural alignment of the productive binding modes for CtNTH/dGMP and AdaV/dAMP, highlighting compared residues 175–177 (green), 197-202 (blue), 271–275 (light yellow), and 300–306 (pink). **c** HPLC analysis of enzymatic reactions catalyzed by CtNTH[H274Y], AdaV[WT], and AdaV[F303V]. The experiments were repeated independently three times with similar results. **d** Steady-state kinetic analysis of CtNTH[WT], CtNTH[H274Y], AdaV[WT], and AdaV[F303V]. n.d.: not determined. Source data are provided in Source Data file.

Structure analysis unveils a stable water-bridged hydrogen bonding interaction between the phenolic hydroxyl group of Tyr274 and the 3′-OH group of dAMP, which may help to stabilize mode 2 (Fig. 6c). The importance of the phenolic hydroxyl group is evidenced by the observation that H274F exhibits a lower activity to dAMP compared with H274Y (Supplementary Fig. 34).

In CtNTH[H274Y]/dGMP system, the H274Y mutation decreases the ratio of binding mode 1, the active binding mode of dGMP, from 65.1% to 20.8%. Furthermore, Tyr274, which is larger than histidine, positions Phe272 to be closer to the binding site of the deoxyribose ring. To avoid steric repulsions with Phe272, the deoxyribose ring moves farther away from the oxo and Cl groups compared to that in the wild-type CtNTH (Fig. 6d). The elongated O···H and C2'···Cl distances ($d_{O·H} = 4.1 \pm 1.0$ Å, $d_{C2'-Cl} = 4.6 \pm 1.1$ Å) indicated that dGMP deviates from the productive state in mode 1, contributing to the decreased activity of CtNTH[H274Y] towards dGMP. In line with these observations, the dynamic cross-correlation analysis (DCCA) (Supplementary Fig. 50, Supplementary Table 6) unveils correlated motions of Tyr274 with Phe272 (correlation coefficient 0.61) and dGMP (correlation coefficient

−0.94), supporting the role of Tyr274 in affecting the binding of substrate.

The AdaV system also showed a similar change in binding mode preference. MD simulations indicated that the F303V mutation shifts the dominant binding mode from mode 2 to mode 1 (Fig. 6b), resulting in higher activity of AdaV[F303V] towards dGMP compared to dAMP. Analysis suggested that F303V mutation may impact the distribution of binding modes by influencing the motion of its neighboring residue Arg302, which forms stable hydrogen bonding interactions with the phosphate group of substrates (Fig. 4, Supplementary Table 5). Being situated at the C-terminal loop, Arg302 is quite flexible, enabling it to approach Arg177 for binding in mode 1 or move closer to Arg237 for binding in mode 2 (Fig. 4). DCCA analysis revealed a strong correlation between the motion of the phosphate group and Arg302 (correlation coefficient: −0.45 to −0.73, Supplementary Table 7), indicating that the binding mode may be influenced by Arg302. Furthermore, the motion of Arg302 is highly correlated with F/V303 (correlation coefficient: 0.58 to 0.87). These findings collectively suggest that F303V likely alters the binding mode by affecting the motion of Arg302. In wild-

**a**

|  |  | CtNTH[WT] | CtNTH[H274Y] |
|---|---|---|---|
| **dAMP** | mode 1 | 74.4% | 32.5% |
|  | mode 2 | 20.4% | 66.6% |
| **dGMP** | mode 1 | 65.1% | 20.8% |
|  | mode 2 | 8.9% | 26.2% |

**b**

|  |  | AdaV[WT] | AdaV[F303V] |
|---|---|---|---|
| **dAMP** | mode 1 | 24.5% | 57.3% |
|  | mode 2 | 73.3% | 33.3% |
| **dGMP** | mode 1 | 28.2% | 65.8% |
|  | mode 2 | 71.1% | 32.8% |

**c**

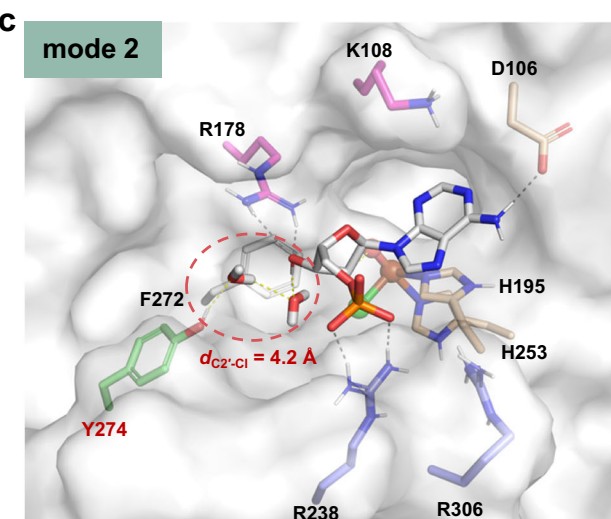

**d**

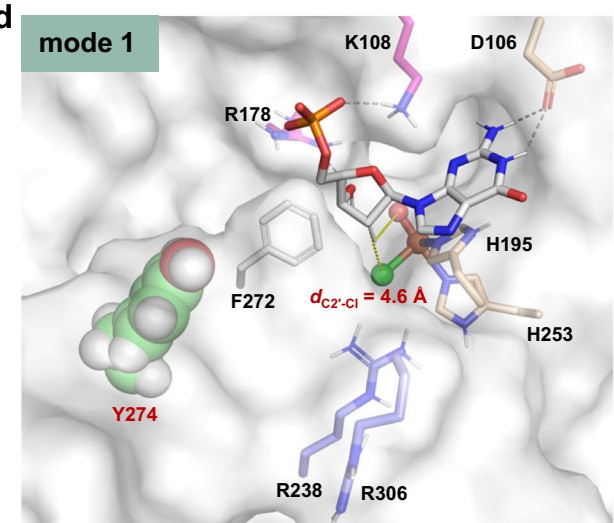

**Fig. 6 | Effects of the second-sphere residues on the binding modes of nucleotide halogenases. a** Percentage of binding modes for CtNTH[WT] and CtNTH[H274Y]. **b** Percentage of binding modes for AdaV[WT] and AdaV[F303V]. **c** Representative structure of binding mode 2 for CtNTH[H274Y]/dAMP; Tyr274 form a stable water-bridged hydrogen bonding interaction with 3'-OH group of dAMP (highlighted in the dotted cycle). **d** Representative structure of binding mode 1 for CtNTH[H274Y]/dGMP; Tyr274 positions Phe272 to be closer to the sugar-binding site, making the deoxyribose ring of dGMP to get farther away from the catalytic iron center.

type AdaV, Phe303 can engage in cation-π stacking interaction with Arg302, as shown in Fig. 4. However, the F303V mutation replaces an aromatic residue with an aliphatic residue, eliminating the cation-π interaction with Arg302 and reducing the size of the sidechain. These changes are expected to affect the motion pattern of Arg302. Above analysis highlights the essential role of C-terminal residues in substrate binding, in line with previous studies of other αKGHs, such as AmbO5, WelO5, HalB, and HalD[19,50].

Altogether, these results conclude that the second-sphere residues confer the nucleotide specificity by regulating the binding mode. Based on the understanding of the substrate selectivity mechanism, we revisited the five candidate enzymes showcased in Fig. 2a and observed conserved Y273 and F303 residues (Supplementary Fig. 1), supporting their classification as dAMP halogenases.

## Discussion

C2'-halogenation represents an essential modification to enhance the drug-like properties of nucleotide analogs and holds a great potential for subsequent transformations. While AdaV has provided a valuable method for the direct C2'-halogenation of dAMP, its narrow substrate scope limits its applications. Here, the discovery of dGMP halogenases, VaNTH and CtNTH, expands the family of the nucleotide halogenase. Biochemical characterization of these enzymes has demonstrated their ability to catalyze bromination and azidation reactions, indicating their promising role as biocatalysts for nucleotide modifications.

Additionally, the proposed nucleotide discrimination mechanism, in which the binding pose of the phosphate group controls the nucleobase specificity, adds a new mechanistic scenario to nucleotide-binding enzymes. More importantly, it has inspired the engineering of

substrate specificity. In this work, engineering second-sphere residues located between the two phosphate-binding sites successfully alters the A/G specificity of nucleotide halogenases. Structural and molecular dynamics analysis revealed that these second-sphere residues regulate the nucleotide specificity by controlling the binding pose of the phosphate group.

Overall, the discovery and nucleotide specificity engineering of nucleotide halogenases strengthen their applicability as tools for nucleotide modification. The novel A/G discrimination mechanism enriches our understanding of the substrate specificity for nucleotide-binding enzymes, providing invaluable insight for enzyme engineering. Further exploration of the potential of these enzymes in biocatalytic or chemoenzymatic synthesis may facilitate the creation of new nucleotide analogs, thereby advancing chemical biology research and drug discovery.

## Methods

### Materials

All chemicals and solvents were purchased from commercial suppliers (Bidepharm, Macklin, and Sangon). DNA polymerase (KOD-Plus-Neo) was purchased from TOYOBO Life Science. DpnI restriction enzyme was purchased from Takara Bio. Halogenases genes and primers were synthesized by GENEWIZ.

### Bioinformatic selection of nucleotide halogenase candidates

To select nucleotide halogenase candidates for functional screening, the protein sequence of reported dAMP halogenase AdaV was used as a query for BLAST against the non-redundant protein database with a cutoff value of 30% sequence identity. BLAST hits were aligned using

multiple sequence alignment in EMBL-EBI (https://www.ebi.ac.uk/Tools/msa/mafft/). The sequence position of the HXD/E or HXG/A motif was identified manually. 10 nucleotide halogenase candidates which contain the characteristic HXG/A motif were obtained.

## Phylogenetic analysis of VaNTH and CtNTH with reported Fe(II)/α-ketoglutarate-dependent halogenases

The protein sequence of indole alkaloids halogenases[18–20,51], dechloroacutumine halogenase[24], nucleotide halogenases[27,39] and amino acids halogenases[25] for phylogenetic analysis was obtained from literature. Halogenase sequences were aligned using multiple sequence alignment in EMBL-EBI (https://www.ebi.ac.uk/Tools/msa/mafft/)[52]. The phylogenetic tree was drawn using iTOL (https://itol.embl.de/)[53].

## Plasmid construction and protein expression

The genes encoding VaNTH and CtNTH were cloned into pET28a between NdeI and XhoI. Each plasmid was then transformed into E. coli BL21 (DE3). Recombinant E. coli BL21 (DE3) including the gene of halogenases were cultivated at 37 °C and 220 rpm in 1 L LB medium containing 50 µg/ml kanamycin. Until $OD_{600}$ reached 0.6–0.8, the cells were induced with a final concentration of 0.2 mM IPTG and further cultivated at 16 °C and 220 rpm for 16 h. The cells were then pelleted by centrifugation at 4000 g and 4 °C for 20 min and the supernatant was discarded. The cell pellet was resuspended in Buffer A (50 mM HEPES, 250 mM NaCl, 20 mM imidazole, pH 7.5) and disrupted with ultrasonication for 30 min at 4 °C (work for 2 s, pause for 3 s, 150 W). The lysate was then centrifuged at 8800 g for 30 min at 4 °C to separate the soluble and insoluble fractions. The supernatant was loaded onto a His-Trap HP nickel affinity column pre-equilibrated with buffer A. The column was first washed with Buffer A, then the target protein was eluted off by buffer B (50 mM HEPES, 250 mM NaCl, 300 mM imidazole, pH 7.5). The collected proteins were verified by SDS-PAGE analysis. The protein was digested by thrombin to remove the N-terminal $His_6$ tag and desalted using centrifugal filter (10 kDa, Millipore) into HEPES buffer (50 mM, pH 7.5). All proteins were aliquoted, flash frozen in liquid nitrogen and stored at −80 °C for further biotransformation reactions.

## Site-directed mutagenesis

Site-directed mutagenesis was performed by whole plasmid PCR, using recombinant pet28a-CtNTH as a template and KOD-Plus-Neo DNA polymerase kit. Primers used in this study are listed in supplementary information. The PCR products were verified by nucleic acid gel and digested with 1 µL of DpnI restriction enzyme for 0.5 h at 37 °C to remove the methylated templates. Then 10 µL digestion mixtures were transformed into E. coli BL21 (DE3). Single colonies were picked up and cultivated on LB medium (50 µg/mL kanamycin) overnight. The plasmids of mutations were confirmed by sequencing.

## In vitro activity assay of enzymes with chloride ion

A 200 µL in vitro assay was performed under the standard reaction: 10 µM protein, 0.5 mM $(NH_4)_2Fe(SO_4)_2$, 2.5 mM α-ketoglutaric acid sodium salt, 1 mM ascorbic acid, 10 mM NaCl and 1 mM substrate in HEPES buffer (pH 7.5). Reactions were initiated by addition of substrates and allowed to proceed in constant temperature oscillator (30 °C, 600 rpm) for 1 h. The reactions were quenched by adding equal volumes of methanol. Samples were then analyzed by HPLC and LC–MS.

## In vitro activity assay with other halide and pseudohalide anions

The reaction system included the following: 30 µM protein, 0.5 mM $(NH_4)_2Fe(SO_4)_2$, 2.5 mM α-ketoglutaric acid sodium salt, 1 mM ascorbic acid, 10 mM NaBr/NaI/NaN₃/NaNO₂/NaOCN/NaSCN and 1 mM dGMP in HEPES buffer (pH 7.5). Halogenases used in these assays were desalted into HEPES buffer (50 mM, pH 7.5), trying to prevent the competition

from the native chloride ion found in the enzyme storage buffer. The reaction systems were performed at 30 °C for 12 h and quenched by adding equal amount of methanol.

## Kinetic parameters determination

To determine the kinetic parameters of VaNTH, CtNTH, AdaV and their variants to dAMP and dGMP substrate. Reactions were initiated by the addition of varying concentrations of substrates (0-1.6 mM). The initial reaction rates of the first 10 min were determined by HPLC analysis. Kinetic parameters were determined from fitting of the initial rate data to the Michaelis–Menten equation.

## HPLC analysis

Samples were analyzed by HPLC (Agilent 1260, USA) using Diamonsil C18 column (DIKMA, China, 4.6 × 250 mm) at 30 °C. An injection volume of 10 µL was always used for all samples. The analytes were eluted using an initial solvent composition of 95% solvent A (0.1% TFA in water) and 5% solvent B (methanol) at a flow rate of 1 mL/min that was held for 2 min. This was followed by a linear gradient to 95% solvent B over 8 min. This solvent composition was held for 2 min before returning to initial conditions. The initial conditions were held for 3 min before the next sample was injected. The elution was monitored by 254 nm UV absorption.

## LC-HRMS analysis

LC-HRMS analysis was carried out on a Shimadzu LC-30AD system with AB SCIEX TripleTOF 6600+ connected to electrospray ionization (ESI) mass spectrometer. Samples were analyzed by LC-HRMS, equipped with a C18 ACQUITY column (Waters, 2.1 × 100 mm) at 30 °C. Analytes were eluted using an initial solvent composition of 95% solvent A (0.1% formic acid) and 5% solvent B (acetonitrile) at a flow rate of 0.3 ml/min. This initial solvent composition was held for 1 min followed by a linear gradient to 95% solvent B over 6 min. This composition was held for another 2 min before returning to initial conditions. An ESI ion source in positive ion mode was used for detection using a 500 °C heater temperature.

## NMR spectroscopy

The chlorinated dGMP was purified from the enzymatic reaction using HPLC with semi-preparative C18 column. NMR spectra were recorded using a Bruker Ascend 500 NMR spectrometer. Data for $^1H$ NMR are reported as follows: chemical shift (ppm), multiplicity (d = doublet, t = triplet, q = quartet), coupling constant (Hz), integration. Data for $^{13}C$ NMR are reported as chemical shift (ppm).

## Protein structure and system preparation

The initial structure of AdaV/dAMP was obtained from a previously reported crystal structure in the Protein Data Bank (PDB: 7V7X). The AdaV/dGMP system was constructed by docking dGMP to the active site of the crystal structure of AdaV (PDB: 7W5T). Given the high sequence identity (51.0%) between CtNTH and AdaV, we utilized Swiss Model to construct the 3D structure of CtNTH by homology modeling[54], using AdaV as a template (PDB ID: 7W5T). We also used Alphafold2 to generate a structural model of CtNTH[55]. Similar structures were obtained from both methods, validating their reliability. The structural model generated by Swiss Model was utilized for the following computational studies. dGMP and dAMP were docked into the active site of CtNTH using Autodock Vina[56] to construct CtNTH/dGMP and CtNTH/dAMP complexes, respectively. Among the generated docking poses, the top-ranked pose, which is similar to the binding pose in the crystal structure of AdaV/dAMP, was used as the initial structure for MD simulations. In all these systems, Fe is replaced by Fe-oxo and α-ketoglutarate is replaced by succinate (Suc)[57,58] to construct the high-valent iron-oxo intermediate which is the active species for the halogenation reaction. The protonation states of

titratable residues were assigned using the H + + webserver with a pH of 7.5[59]. Specifically, the protonation states of residues in the active site were checked by visual inspection. The constructed protein-substrate structures of the wild-type CtNTH and AdaV were used to generate the corresponding protein-substrate complex structures for the variant CtNTH[H274Y] and AdaV[F303V], respectively. Mutations (H274Y for CtNTH[H274Y] and F303V for AdaV[F303V]) were introduced using PyMOL.

Parameters for the succinate ligand were generated by the Antechamber22 module in AMBER20 package using the general AMBER force field (GAFF2)[60], with partial charges set to fit the electrostatic potential generated at the HF/6-31 G(d) level by the RESP model[61]. Parameters for the iron coordinated cluster were generated using Metal Centre Parameter Builder tool (MCPB) at the UB3LYP-D3 level with the 6-31 G* basis set for C, H, O, N, Cl atoms and LANL2DZ basis set for the iron atom[62]. Nucleic acid force field bsc1[63] was used for dAMP and dGMP, with partial charges set to fit the electrostatic potential generated at the HF/6-31 G(d) level by the RESP model. The AMBER ff14SB force field was used for all standard protein residues[64,65]. Topology and coordinate files for the constructed structures were prepared in AMBER20 using the tleap utility. The protein−substrate complexes were solvated in a periodic rectangular prism box with at least a 10 Å buffer of TIP3P[66] water and neutralized with Na$^+$ and Cl$^-$ counterions.

## Classical MD simulations

MD simulations were performed using the AMBER20 package with the GPU accelerated PMEMD code[67]. Equilibration was carried out with the following protocol: (i) 10,000 steps energy minimization of solvent and 50,000 steps energy minimization of the protein and substrate, (ii) The system was heated up to 300 K over 0.1 ns with harmonic constraints of 10 kcal/mol, under the canonical ensemble (NVT) conditions, (iii) A 500 ps simulation was conducted under the isothermal-isobaric ensemble (NPT) conditions with applied constraints gradually reduced from 10 to 0 kcal/mol. Following equilibration, a 100 ns restrained MD simulation was run at constant temperature (300 K) and pressure (1 atm) by NPT simulations. Distance restraints was imposed on C2′···Cl and O···H using a harmonic potential with a force constant of 50 kcal · mol$^{-1}$·Å$^{-2}$ (refer to Supplementary Table 8). Subsequently, a 400 ns unrestrained productive MD simulation was performed at a constant temperature of 300 K and pressure of 1 atm. The integration time step of the simulations was set to 2 fs, and the nonbonded cutoff length was set to 10 Å. The thermostat and barostat control were used with Berendsen pressure compressibility at 4.57E-5 bar$^{-1}$ and Berendsen pressure relaxation time at 100 fs. The periodic boundary conditions were used with the Particle-Mesh Ewald method for electrostatic interactions. Three replicates of MD simulations were conducted for each system, and all productive trajectories were combined for further analysis.

## Analysis of MD trajectories

Trajectory clustering was performed based on the substrate binding mode (Mode 1, Mode 2, and other modes) using the specified criteria. Mode 1 entails the phosphate group of the nucleotide forming at least one hydrogen bond with Arg178 (for CtNTH and CtNTH[H274Y]) or Arg177 (for AdaV and AdaV[F303V]). Mode 2 involves the phosphate group of the nucleotide forming at least one hydrogen bond with Arg238 (for CtNTH and CtNTHH[274Y]) or Arg237 (for AdaV and AdaV[F303V]). For other modes, the phosphate group of the nucleotide does not form any hydrogen bond with Arg178 and Arg238 (for CtNTH and CtNTH[H274Y]) or with Arg177 and Arg237 (for AdaV and AdaV[F303V]). Hydrogen bond interactions between the nucleotide substrate and enzyme were analyzed using CPPTraj[68] module from Ambertools utilities with a hydrogen bond distance cutoff of 3.2 Å[69]. Representative structures of modes 1 and 2 were identified by applying the density peak clustering algorithm[70] to determine the cluster center based on the RMSD of the core active site, comprising the key residues, iron-oxo center and substrate. For instance, the active site of CtNTH system used in the calculation includes Asp106, Lys108, Arg178, His193, His195, Arg238, His253, Phe272, His274, Arg306, the iron-oxo center, and the substrate. Dynamic cross correlation and principal component analyses were performed with Bio3d using the backbone Cα atoms of the production dynamics trajectory[71–73].

## Reporting summary

Further information on research design is available in the Nature Portfolio Reporting Summary linked to this article.

## Data availability

The data necessary to support the findings of this study are available within the main text and supplementary information. Sequence data for enzymes used in this study are available online: VaNTH (NCBI Reference Sequence: WP_204007738.1), CtNTH (WP_217394979.1). Source data are provided with this paper.

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

## Acknowledgements

The authors gratefully thank the financial support from the Shenzhen Science and Technology Program (Grant No. RCYX20200714114736199), the National Natural Science Foundation of China (no. 22173077), the Guangdong Basic and Applied Basic Research Foundation (no. 2023B1515020052), Warshel Institute for Computational Biology funding from Shenzhen City and Longgang District, and the Ganghong Young Scholar Development Fund. The authors appreciate the anonymous reviewers' valuable and insightful comments.

## Author contributions

J. Ni conducted the bioinformatics analysis, enzyme characterization and engineering experiments, and MD simulations. J. Zhuang and Y.-C. Chiang contributed to the MD trajectory analysis. Y. Shi participated in the enzyme characterization experiments. G.-J. Cheng designed and oversaw the project. J. Ni and G.-J. Cheng wrote the manuscript.

## Competing interests

The authors declare no competing interests.
