## [Peer Review File · Nature Communications]

Discovery and substrate specificity engineering of nucleotide halogenasesREVIEWER COMMENTS

Reviewer #1 (Remarks to the Author):

In this manuscript, the authors have implemented experimental and computational methods to identify nucleotide-specific halogenases. The authors have isolated two halogenases, VaNTH and CtNTH. AdaV halogenase, specific to the dAMP substrate, has been used to identify these halogenases using BLAST homology search. The authors have also determined that these halogenases are specific to dGMP substrate. Using Molecular dynamics simulations, the authors have identified the important second coordination residues important to substrate selectivity. Overall, this study gives valuable insights into the importance of the second coordination sphere residues in the reactivity of 2OG-dependent halogenases. The work is significant in the fields subject to a major revision in the following directions.

Review Questions:

- 1) The MD simulations are relatively short – 500ns. Did the authors use multiple runs or only single runs?
- 2) How the authors decided to rule out eight enzymes that have high sequence similarity to AdaV. Clarification is needed on the logical reasoning behind the exclusion. Just the sequence similarity alone cannot be used to predict their substrate selectivity. The authors have to explain this better.
- 3) It is well known that Second coordination sphere residues have distinct roles in the substrate binding and catalytic activity. However, it would be interesting to understand the authors' reasoning for narrowing down their mutational analysis to two SCS residues, V273 and H274, for switching the substrate specificity.
- 4) Some relevant publications on halogenases should be included e.g. ACS Omega 2018, 3, 5, 4847–4859; Scientific Reports volume 7, Article number: 17395 (2017); also on Fe(II)/2OG enzymes: <https://pubs.acs.org/doi/full/10.1021/jacsau.2c00345>; <https://pubs.acs.org/doi/abs/10.1021/acscatal.2c00024>
The authors should perform DCCA to explore the correlated motions to some key simulations. The references suggested above in point 4 are good guidance.

Reviewer #2 (Remarks to the Author):

The authors describe the discovery of two novel Fe(II)/αKG-dependent halogenases (VtNTH and CtNTH) that catalyze the chlorination of 2'-deoxyguanosine monophosphate (dGMP). The enzymes from *Catellatospora tritici* and *Virgisporangium aurantiacum*, which were identified via a homology-based search, were functionally expressed in *E. coli* and showed promising halogenation activity in vitro. While the enzymes exhibited a narrow substrate scope, the authors gave evidence that alternative anions, including Br⁻ and N₃⁻ could be incorporated into the enzymes' native substrate dGMP. Intrigued by the tight nucleotide specificity shown by the nucleotide halogenases (CtNTH, VtNTH and AdaV), the authors used molecular dynamics simulations with complementary mutagenesis experiments to elucidate the structural elements governing nucleotide specificity.

From my point of view, the findings presented in this manuscript make a nice contribution to the field as the discovery of new freestanding Fe(II)/αKG-dependent halogenases are rare. This work further contributes to our understanding of factors governing nucleotide specificity of these enzymes. However, I regret to say that I do not believe the findings presented here reach the level of impact required for publication in *Nature Communications* – particularly given the prevailing structural insights that exist in the literature for AdaV.

I have highlighted more specific concerns below, which might help to improve the manuscript should the authors decide to submit it elsewhere. In addition, I found the manuscript to be quite difficult to read at times and careful revisions of English and phrasing is required.

- The authors provide app kcat values for the wild-type CtNTH and its single mutant (H274Y) towards both dGMP and dAMP. How do these values compare to the wild-type AdaV towards its natural substrate (dAMP)? App kcat values would also be valuable for the AdaV_A272V_Y273H variant towards both substrates.

- The authors state that single and double mutants of CtNTH and AdaV were created and their activities towards dGMP and dAMP were assessed experimentally. Only the variants that showed the desired switch in activity are discussed - but what about the others? A more critical assessment of these variants is warranted.
- The authors have characterized chlorinated dGMP by NMR. Can an isolated yield be provided?
- The authors demonstrate that the newly discovered CtNTH exhibits anion promiscuity, for example, Br⁻ and N₃⁻ can also be incorporated into dGMP. The authors should acknowledge here that these findings are consistent with results found for other α-KGHs. For example, WelO5* and SaDAH are also capable of installing anions other than Cl⁻. It would also be beneficial to state in the main text which anion yielded the best transformation results. The corresponding supplementary figure (S6) should also be mentioned in the main text – this is something which is generally often neglected throughout the manuscript.
- Given that AdaV does not show hydroxylation activity unless the enzyme has been engineered (AdaVQ203A/V269A, Zhai et al, ACS Catal. 2022, 12, 22, 13910–13920) the authors should state more clearly that low levels of the hydroxylated product of dGMP can be generated by CtNTH, even in the presence of trace amounts of Cl⁻, possibly originating from the enzyme storage buffer. While this is conveyed well in the supplementary information, the main text would benefit from an additional sentence with this information before discussing the effects of OCN⁻ or SCN⁻ on hydroxylation activity.
- The authors state that AdaV “specifically” acts on dAMP, however it is known that the enzyme can also halogenate 2'-ddAMP and 2'-dIMP. It would be good to highlight this or mention that AdaV preferentially accepts dAMP as a substrate.
- The authors state that the initial CtNTH/dGMP structure for MD simulation was selected based on the results from mutagenesis experiments. However, mutagenesis experiments are not described until later in the text. I found this to be confusing and the authors should provide a clearer rationale for their decision in the main text.
- Zhao et al had previously indicated that the 5'-phosphate moiety in the 2'-deoxyadenosine scaffold is essential for enzymatic activity. Given their findings, the authors should acknowledge this in the main text.
- Figure 3a and 3b: For completeness, please include R99 and the second iron coordinating histidine, H253, in the relevant figures. Please note that H253 should be shown in all figures presented in the manuscript and supplementary information.
- It's not clear from the manuscript that the activity of AdaV towards dGMP was experimentally tested in this study. The corresponding supplementary figure should be mentioned in brackets here.

Supplementary Information:

- All HPLC chromatograms should include a scale.
- Figure S2_ Please revise the figure legend and correct numbering.

Editorial comments:

Page 1, line 1

The authors should consider changing the title to “Discovery and Substrate Specificity Engineering of Nucleotide Halogenases”.

Page 1, line 13

“The direct C2'-halogenation of the nucleotide 2'-deoxyadenosine-5'-monophosphate (2'dAMP) has recently been achieved using the Fe(II)/α-ketoglutarate-dependent nucleotide halogenase AdaV.”

Page 1, line 15

"However, the limited substrate scope of this enzyme hampers its broader applications."

Page 1, line 17

"In this study, we report two novel halogenases capable of halogenating 2'-deoxyguanosine monophosphate (dGMP), thereby expanding the family of nucleotide halogenases."

Page 1, line 19

"Computational studies reveal that nucleotide specificity is regulated by the binding pose of the phosphate group. Based on these findings, we successfully engineered the substrate specificity of these halogenases by mutating second-sphere residues."

Page 1, line 22

"This work expands the toolbox of nucleotide halogenases and provides a new strategy for nucleotide specificity engineering."

Page 1, line 27

"Late-stage functionalization of C-H bonds has proven to be an efficient method for modifying complex molecules."

Page 2, line 47

"The recent discovery of the Fe(II)/ α -ketoglutarate-dependent radical nucleotide halogenases, AdaV, provides a late-stage sp³ C-H halogenation method that enables direct modification of the deoxyribose ring at the C2' position with high regio- and stereoselectivity (Fig. 1c)."

Page 2, line 54

"To address this limitation, new α KGHs that can act on alternative nucleotide substrates are highly sought after."

Page 2, line 55

The authors should consider removing this sentence as it implies that a more a more comprehensive engineering campaign was carried to expand the substrate scope of these enzymes.

Page 2, line 57

"In this work, we reported the discovery and characterization of two novel Fe(II)/ α -ketoglutarate-dependent halogenases, which are capable of halogenating dGMP."

Page 3, line 60

"Based on the nucleotide discrimination mechanism, we successful engineered switched their substrate specificity."

Page 3, line 68

"To search for potential nucleotide halogenases candidates, we performed a sequence-based BLAST (basic local alignment search tool) analysis in NCBI (national center for biotechnology information) using AdaV as the query."

Page 5, line 112

"Overall, through bioinformatic and biochemical analysis, we have identified two new dGMP halogenases that also display bromination and azidation activities."

Page 7, line 149

"methylene linker"

Page 9, 194

"The divergent, productive, binding modes observed in AdaV/dAMP and CtNTH/dGMP systems can be explained by the fact that the amino groups of the adenine and guanidine moieties are located at different positions on the purine ring, C6 and C2 respectively."

Page 11, line 224

Please re-phrase this sentence. In its current form it's not coherent what the authors are trying to convey.

“While AdaV has provided a valuable method for the direct C2'-halogenation of dAMP, its narrow substrate scope limits its applications.”

Reviewer #3 (Remarks to the Author):

Ni et al report a study of an interesting family of Fe/2OG radical halogenases that utilize nucleotide substrates that was first established upon the discovery of a family member that utilizes dAMP as a substrate. They characterize the additional family members and find that dGMP is preferred for a subset, which I find to be quite interesting. They utilize MD studies to generate a hypothesis for the mode of binding of the substrate relative to the Fe center, yielding a reasonable structure that shows potential fluctuation between two nucleotide conformations. However, I do not have the expertise or training to critically judge the MD simulations. By use of sequence alignment, they identify residues in the second sphere that differ between the dAMP and dGMP halogenases, which are proposed to control the substrate selectivity. They test this hypothesis by swapping the residues in the dGMP halogenase for the corresponding amino acids found in the dAMP halogenase. They further utilize MD to explore the change in selectivity of their dGMP halogenase to favor dAMP and note that the population of the two possible nucleotide conformations has shifted.

Overall I think that this paper is interesting as it identifies new substrates for this family of radical halogenases and gives information on the nucleotide binding selectivity, which could be valuable for biocatalytic applications of these enzymes. However, the claims are too strongly worded for me throughout the engineering and mechanistic sections. I believe that these data can be described more thoughtfully without altering the impact of the paper but I am uncomfortable with some of the conclusions as written.

1. The claim is made that a halogenase with swapped selectivity has been engineered. I apologize if I have missed some data in the SI, however the data in Figure 5 shows kinetics for the single mutant and then endpoint analysis for the double mutant. The kinetic data shows that the H274Y mutant shows a 20 fold decrease in k_{cat} for dGMP and a 2 fold increase in k_{cat} for dAMP. However, the k_{cat} for dAMP remains almost an order of magnitude below that for the wild-type enzyme for dGMP, so I do not believe one can say that the selectivity is swapped. From this data, I feel that one can only conclude that H274 appears to participate in substrate selectivity and when it is mutated to Y that both activity and selectivity is lost. Second, there is no full steady-state analysis of either H274Y or the double mutant. For the double mutant, I only see endpoint analysis. Both of these mutants need to have full steady-state analysis provided. Currently, it is impossible to assess the success of the swap and whether it is mostly derived from a catalytic defect that more strongly affects dGMP compared to dAMP because dGMP is the native substrate (a common finding).

2. The claim is made that: “Altogether, these results conclude that the second-sphere residues confer the nucleotide specificity by regulating the binding mode, which in turn validates our engineering strategy.” This claim is also very strongly worded when there is very little experimental support for this claim given that it is based on simulation as well as incompletely characterized mutants that do not necessarily clearly show a mechanistically valid swap in selectivity. I think this can be reworded easily to say something more along the lines that these results suggest that nucleotide specificity is regulated by the binding mode. However, I am not sure if it also validates the engineering strategy.

Reviewer #1 (Remarks to the Author):

In this manuscript, the authors have implemented experimental and computational methods to identify nucleotide-specific halogenases. The authors have isolated two halogenases, VaNTH and CtNTH. AdaV halogenase, specific to the dAMP substrate, has been used to identify these halogenases using BLAST homology search. The authors have also determined that these halogenases are specific to dGMP substrate. Using Molecular dynamics simulations, the authors have identified the important second coordination residues important to substrate selectivity. Overall, this study gives valuable insights into the importance of the second coordination sphere residues in the reactivity of 2OG-dependent halogenases. The work is significant in the fields subject to a major revision in the following directions.

Response: Thank you for the positive comments.

Review Questions:

1) The MD simulations are relatively short – 500ns. Did the authors used multiple runs or only single runs?

Response: We conducted three independent 500ns MD simulations for each system and combined the productive trajectories from these replicates for our analysis. Considering that the proteins under study did not exhibit significant conformational changes, three replicates of 500ns MD simulations should be adequate to sample substrate binding poses effectively.

2) How the authors decided to rule out eight enzymes that have high sequence similarity to AdaV. Clarification is needed on the logical reasoning behind the exclusion. Just the sequence similarity alone cannot be used to predict their substrate selectivity. The authors have to explain this better.

Response: This is an excellent question, and we appreciate the opportunity to provide further clarification. Initially, the exclusion of the eight enzymes was based on their sequence identity. As detailed in Table S1, previous research (*ACS Catal.* 2022, 12, 22, 13910–13920) identified the enzymes #2, #5, and #8 as dAMP halogenases. Additionally, the remaining five enzymes (#1, #3, #4, #6, and #7) exhibited over 72% sequence identity with these characterized dAMP halogenases, suggesting a strong likelihood of sharing the same substrate specificity. Consequently, we chose to focus on the enzymes listed at the bottom of the table, as they displayed lower sequence identity (55.5% and 53.7%) with the reported dAMP halogenase, indicating a potential for a different substrate scope.

Table S1. Percent identity matrix of AdaV analogs.

	AdaV*	1	2*	3	4	5*	6	7	8*	9	10
AdaV *	100	86.7	85.5	86.3	86.8	73.8	75.2	70.0	67.0	55.5	51.0
MDF2704590.1 (1)	86.7	100	88.2	87.4	85.5	74.5	76.0	69.4	69.7	55.7	52.2
WP_182876399.1 (2) * (DI270)	85.5	88.2	100	89.5	84.7	75.5	75.6	70.6	65.9	54.5	50.7

WP_184541559.1 (3)	86.3	87.4	89.5	100	87.4	76.5	75.7	70.9	68.0	55.1	52.6
WP_225289932.1 (4)	86.8	85.5	84.7	87.4	100	72.0	74.2	69.9	65.0	53.2	51.9
WP_030883106.1 (5) * (ADK38)	73.8	74.5	75.5	76.5	72.0	100	72.6	72.7	69.8	54.2	51.4
WP_279554523.1 (6)	75.2	76.0	75.6	75.7	74.2	72.6	100	80.9	67.7	56.1	53.3
WP_189993169.1 (7)	70.0	69.4	70.6	70.9	69.9	72.7	80.9	100	66.4	54.9	51.0
WP_158075676.1 (8) * (BJP25)	67.0	69.7	65.9	68.0	65.0	69.8	67.7	66.4	100	54.7	53.7
WP_204007738.1 (9) (VaNTH)	55.5	55.7	54.5	55.1	53.2	54.2	56.1	54.9	54.7	100	67.3
WP_217394979.1 (10) (CtNTH)	51.0	52.2	50.7	52.6	51.9	51.4	53.3	51.0	53.7	67.3	100

As correctly noted by the reviewer, the sequence similarity alone cannot reliably predict substrate selectivity. The final decision to exclude these enzymes was guided by our understanding of the substrate selectivity mechanism. Our combined computational and experimental analysis revealed the essential role of His274 in nucleotide selectivity of dAMP halogenases. However, the five enzymes (#1, #3, #4, #6, and #7) feature with a conserved tyrosine at the corresponding site, aligning with that of dAMP halogenases (Figure S1), providing additional evidence supporting their classification as dAMP halogenases.

In response to the reviewer's insightful comment, we have modified the manuscript to better explain the logical reasoning behind the exclusion.

Page 3, line 71:

“The generated ten candidates include three recently identified dAMP halogenases³⁹ (DI270, ADK38, and BJP25 in Fig. 2a and Table S1), which were thus excluded. The remaining seven candidates were submitted for maximum-likelihood phylogenetic analysis with reported free-standing α KGHs^{18-20, 24, 25, 27, 39, 40}. As depicted in Fig. 2a, five candidates (labeled as 1, 3, 4, 6, and 7), which share over 70% sequence identity with AdaV, clustered within the same clade as the characterized dAMP halogenases. This clustering strongly suggests their classification as dAMP halogenases. Intriguingly, the last two candidates (9 and 10) formed a distinct clade in the constructed phylogenetic tree, indicative of their potential as nucleotide halogenases with differing substrate specificity.”

Page 16, line 359:

“Based on the understanding of the substrate selectivity mechanism, we revisited the five candidate enzymes showcased in Fig. 2a and observed conserved Y273 and F303 residues (Fig. S1), supporting their classification as dAMP halogenases.”

3) It is well known that Second coordination sphere residues have distinct roles in the substrate binding and catalytic activity. However, it would be interesting to understand the authors' reasoning for narrowing down their mutational analysis to two SCS residues, V273 and H274, for switching the substrate specificity.

Response: Thank you for your valuable comments. In our study, we compared the residues located between the two phosphate-binding sites for dAMP and dGMP halogenases in the active binding modes. Specifically, we focused on residues that distinguish between dGMP and dAMP halogenases while remaining conserved within each enzyme subfamily. Based on this criterion, we identified six residues and subjected them for mutational analysis to pinpoint the specific residues regulating the substrate specificity. The detailed explanations have been incorporated into the revised manuscript and quoted below.

Page 11, line 248:

“However, our simulations show that the nucleotide specificity of CtNTH and AdaV is regulated by the binding pose of the phosphate group. This insight led us to explore the potential for engineering nucleotide specificity by mutating residues located between the two phosphate-binding sites, which could potentially influence the binding modes distribution.

To this end, we aligned the sequences of new dGMP halogenases with the reported dAMP halogenases to compare the residues located between the two phosphate-binding sites (Fig. 5a). As highlighted in Fig. 5b, these residues are located at the C-terminal α -helix, three anti-parallel β -sheets (β 4, β 6, and β 11), and their connecting loops. Among these residues, we specifically focused on the sites that distinguish between dGMP and dAMP halogenases while remaining conserved within each enzyme subfamily. Following this criterion, we identified six residues in CtNTH: Ile176, Val273, His274, Val301, Gly304, and Leu305, which correspond to Asn175, Ala272, Tyr273, Ile299, Arg302, and Phe303, respectively, in AdaV (Fig. 5b).

To pinpoint the specific residues involved in substrate selectivity, we constructed six single mutants (I176N, V273A, H274Y, V301I, G304R, and L305F) for CtNTH. These mutations were designed to replace the original residues in CtNTH with the corresponding amino acids found in the dAMP halogenases. The endpoint analysis suggests that I176N, V273A, V301I, G304R, and L305F do not affect the substrate specificity as they still exhibit clear preference towards dGMP (Fig. S33). Intriguingly, H274Y alters the substrate preference of CtNTH, showing increased activity towards dAMP compared to dGMP (Fig. 5c).”

4) Some relevant publications on halogenases should be included e.g. ACS Omega 2018, 3, 5, 4847–4859; Scientific Reports volume 7, Article number: 17395 (2017); also on Fe(II)/2OG enzymes: <https://pubs.acs.org/doi/full/10.1021/jacsau.2c00345>; <https://pubs.acs.org/doi/abs/10.1021/acscatal.2c00024>. The authors should perform DCCA to explore the correlated motions to some key simulations. The references suggested above in point 4 are good guidance.

Response: Thank you for the suggestions. We have cited the mentioned publications in the revised manuscript and found them to be inspiring and useful (ref. 4, 5, 72, 73). Following the suggested references, we performed DCCA to explore the correlated motions to key simulations, as detailed in Figure S50, S51 and Table S6, S7. DCCA analysis revealed correlated motions between the essential second-sphere residues (274,

303) and the phosphate group of substrates, providing supports for the conclusion that second-sphere residues regulate the nucleotide specificity by controlling the binding pose of the phosphate group. Furthermore, the DCCA analysis allowed us to identify correlated motions between the essential second-sphere residues (274, 303) and the residues (F272, R302) interacting with the substrate, thereby offering valuable insights into how they affect the substrate binding pose. These analyses have been incorporated into the revised manuscript.

Thank you very much for all the comments. They were very useful and helpful for us to improve our study.

Reviewer #2 (Remarks to the Author):

The authors describe the discovery of two novel Fe(II)/ α KG-dependent halogenases (VtNTH and CtNTH) that catalyze the chlorination of 2'-deoxyguanosine monophosphate (dGMP). The enzymes from *Catellatospora tritici* and *Virgisporangium aurantiacum*, which were identified via a homology-based search, were functionally expressed in *E. coli* and showed promising halogenation activity in vitro. While the enzymes exhibited a narrow substrate scope, the authors gave evidence that alternative anions, including Br⁻ and N₃⁻ could be incorporated into the enzymes' native substrate dGMP. Intrigued by the tight nucleotide specificity shown by the nucleotide halogenases (CtNTH, VtNTH and AdaV), the authors used molecular dynamics simulations with complementary mutagenesis experiments to elucidate the structural elements governing nucleotide specificity.

From my point of view, the findings presented in this manuscript make a nice contribution to the field as the discovery of new freestanding Fe(II)/ α KG-dependent halogenases are rare. This work further contributes to our understanding of factors governing nucleotide specificity of these enzymes.

Response: Thank you for the positive comments.

However, I regret to say that I do not believe the findings presented here reach the level of impact required for publication in *Nature Communications* – particularly given the prevailing structural insights that exist in the literature for AdaV.

Response: We respectfully disagree with the notion that the prevailing structural insights available in the literature for AdaV diminish the significance of our findings. Our work uncovers a novel mechanism whereby the binding pose of the phosphate group regulates nucleotide specificity. This mechanism cannot be elucidated based on the static crystal structure of the AdaV/dAMP reported in literature. Our study contributes to a deeper understanding of nucleotide halogenases, which could be valuable for biocatalytic applications of these enzymes. Furthermore, the proposed mechanism presents a new model for controlling nucleotide specificity, which has not been reported in the context of nucleotide-binding enzymes before. On the other hand, as recognized by the reviewer, this work also makes a nice contribution to the field as the discovery of new dGMP halogenases expands the family of freestanding α KGHs. We believe that these findings are valuable and merit publication in *Nature Communications*.

I have highlighted more specific concerns below, which might help to improve the manuscript should the authors decide to submit it elsewhere. In addition, I found the manuscript to be quite difficult to read at times and careful revisions of English and phrasing is required.

Response: We have carefully addressed the concerns highlighted below and made revisions to the manuscript to improve its readability and clarity. We appreciate your suggestions, which are instrumental in enhancing the quality of our manuscript.

1.- The authors provide app kcat values for the wild-type CtNTH and its single mutant

(H274Y) towards both dGMP and dAMP. How do these values compare to the wild-type AdaV towards its natural substrate (dAMP)? App k_{cat} values would also be valuable for the AdaV_A272V_Y273H variant towards both substrates.

Response: (1) Thanks for your comment. Following your and reviewer 3's suggestion, we have conducted full steady-state analysis to determine the kinetic parameters for CtNTH^{WT} and CtNTH^{H274Y} towards both dGMP and dAMP. The table below presents a comparison of these values with the wild-type AdaV towards its natural substrate (dAMP):

	AdaV	CtNTH ^{WT}		CtNTH ^{H274Y}	
	dAMP	dGMP	dAMP	dGMP	dAMP
k_{cat}/K_m ($\text{mM}^{-1}\text{min}^{-1}$)	0.44 ± 0.1	17.8 ± 0.1	0.87 ± 0.09	0.47 ± 0.08	1.83 ± 0.06

The updated data reveal that the catalytic efficiency of wild-type CtNTH is higher than AdaV. The catalytic efficiency of CtNTH^{H274Y} toward dAMP is 4-fold that of AdaV.

(2) We apologize that we made a mistake with AdaV^{A272V_Y273H} variant in the previous manuscript. It did not change the nucleotide specificity. With further mutagenesis experiments, we have confirmed that the second-sphere residue F303, located at the C-terminal, plays a role in the substrate preference for AdaV. Specifically, the F303V variant has been found to alter the substrate preference, and its kinetic parameters (k_{cat} , K_m , and k_{cat}/K_m) towards both substrates have been determined. Importantly, this correction does not impact our original conclusion. These updated insights have been included in our revised manuscript.

	1 (dGMP)			3 (dAMP)		
	k_{cat} (min^{-1})	K_m (mM)	k_{cat}/K_m ($\text{mM}^{-1}\text{min}^{-1}$)	k_{cat} (min^{-1})	K_m (mM)	k_{cat}/K_m ($\text{mM}^{-1}\text{min}^{-1}$)
AdaV ^{WT}	n.d.	n.d.	n.d.	0.69 ± 0.03	1.56 ± 0.2	0.44 ± 0.1
AdaV ^{F303V}	0.79 ± 0.05	0.20 ± 0.02	4.0 ± 0.1	0.54 ± 0.02	0.24 ± 0.02	2.30 ± 0.05

2.- The authors state that single and double mutants of CtNTH and AdaV were created and their activities towards dGMP and dAMP were assessed experimentally. Only the variants that showed the desired switch in activity are discussed - but what about the others? A more critical assessment of these variants is warranted.

Response: Thank you for your feedback. We appreciate the reviewer's suggestion and have now included descriptions of these variants in the revised version:

Page 12, line 270:

“To pinpoint the specific residues involved in substrate selectivity, we constructed six single mutants (I176N, V273A, H274Y, V301I, G304R, and L305F) for CtNTH. These mutations were designed to replace the original residues in CtNTH with the corresponding amino acids found in the dAMP halogenases. The endpoint analysis suggests that I176N, V273A, V301I, G304R, and L305F do not affect the substrate specificity as they still exhibit clear preference towards dGMP (Fig. S33). Intriguingly,

H274Y alters the substrate preference of CtNTH, showing increased activity towards dAMP compared to dGMP (Fig. 5c). This suggests that His274 plays a role in substrate selectivity.”

Page 13, line 289:

“We conducted similar single mutation experiments to introduce the corresponding amino acids found in dGMP halogenases into AdaV. It was found that AdaV is highly sensitive to the mutation. These designed mutants (N175I, A272V, Y273H, I299V, R302G, and F303L) significantly diminish the halogenation activity towards dAMP (Fig. S37). Double mutants, R302G_F303L and A272V_Y273H, also eliminate the enzymatic activity (Fig. S38). All these mutations abolish the activity towards dGMP, except for F303L, which yields trace halogenation product.”

3- The authors have characterized chlorinated dGMP by NMR. Can an isolated yield be provided?

Response: We regret that we are unable to provide the precise isolated yield due to the loss of product during the isolation process. Our isolation method involved concentrating the reaction mixture using vacuum freeze-drying equipment, followed by separation using semi-preparative HPLC to collect the product, which was further evaporated to obtain the product powder. Unfortunately, a portion of the product was lost during the HPLC separation step due to the manual collection based on retention time, which is less precise than automatic collection using a preparative HPLC.

Determining the precise isolated yield is currently challenging for us as we do not have preparative HPLC. Additionally, the small-scale nature of the reaction makes it difficult to collect a large amount of product. In our experiment, we only obtained a small amount of product that adhered tightly to the wall of the flask in the last evaporation step. To minimize unnecessary loss, we did not weigh the purified product and instead directly dissolved it in DMSO for NMR analysis.

4- The authors demonstrate that the newly discovered CtNTH exhibits anion promiscuity, for example, Br⁻ and N₃⁻ can also be incorporated into dGMP. The authors should acknowledge here that these findings are consistent with results found for other α -KGHs. For example, WelO5* and SaDAH are also capable of installing anions other than Cl⁻. It would also be beneficial to state in the main text which anion yielded the best transformation results. The corresponding supplementary figure (S6) should also be mentioned in the main text – this is something which is generally often neglected throughout the manuscript.

Response: Following the reviewer’s suggestion, we have incorporated the suggested acknowledgment and citation of Figure S6 into the revised main text:

Page 5, line 107:

“It was found that I^- and NO_2^- were unreactive. Br^- and N_3^- were successfully incorporated to the substrate with inferior reactivity compared to Cl^- (Fig. S13-14). These findings are consistent with the anion promiscuity reported for other α KGHs, such as SaDAH²⁴ and BesD²⁵.”

We appreciate the valuable feedback and have ensured that the corresponding supplementary figure is appropriately mentioned in the main text. Thank you for your input.

5- Given that AdaV does not show hydroxylation activity unless the enzyme has been engineered (AdaVQ203A/V269A, Zhai et al, ACS Catal. 2022, 12, 22, 13910–13920) the authors should state more clearly that low levels of the hydroxylated product of dGMP can be generated by CtNTH, even in the presence of trace amounts of Cl^- , possibly originating from the enzyme storage buffer. While this is conveyed well in the supplementary information, the main text would benefit from an additional sentence with this information before discussing the effects of OCN^- or SCN^- on hydroxylation activity.

Response: Following the reviewer’s suggestion, we have included additional statements in the revised manuscript. We appreciate the valuable insight, and we are glad to have the opportunity to enhance the clarity of our manuscript. Thank you for the suggestion.

Page 5, line 111:

“In all these reactions, the hydroxylation of dGMP was observed as a background reaction, which led to a trace amount of GMP byproduct. It is intriguing that the addition of OCN^- or SCN^- significantly increased the efficiency of hydroxylation reaction (Fig. S8).”

6- The authors state that AdaV “specifically” acts on dAMP, however it is known that the enzyme can also halogenate 2'-ddAMP and 2'-dIMP. It would be good to highlight this or mention that AdaV preferentially accepts dAMP as a substrate.

Response: The statement has been modified.

Page 2, line 51:

“However, the AdaV preferentially accepts dAMP as a substrate²⁷, limiting its applications as a nucleotide halogenation tool.”

7- The authors state that the initial CtNTH/dGMP structure for MD simulation was selected based on the results from mutagenesis experiments. However, mutagenesis experiments are not described until later in the text. I found this to be confusing and the authors should provide a clearer rationale for their decision in the main text.

Response: Thank you for pointing out the confusion regarding the rationale for selecting the initial CtNTH/dGMP structure for MD simulation. The decision was based on the following reasons: (1) The pose chosen (cf1) is the top-ranked docking pose; (2)

It is similar to the binding mode in the crystal structure of AdaV/dAMP; (3) Another representative binding pose (cf2) is not consistent with the mutagenesis experiment that D106A totally abolish all chlorination activity since there is no interaction between D106 and dGMP in cf2. While cf1 is consistent with the D106A mutagenesis experiment.

We agree with the reviewer that our previous statement is confusing as mutagenesis experiments are not described until later in the text. Thus, we modified the sentence as shown below.

Page 6, line 130:

“For CtNTH/dGMP system, the 3D structure of CtNTH was built by homology modeling using AdaV as a template, and dGMP was docked into the active site of CtNTH. The top-ranked pose, which is similar to the binding pose in the crystal structure of AdaV/dAMP, was used as the initial structure for MD simulations (Fig. S21)”

8- Zhao et al had previously indicated that the 5'-phosphate moiety in the 2'-deoxyadenosine scaffold is essential for enzymatic activity. Given their findings, the authors should acknowledge this in the main text.

Response: Thanks for your suggestion. Following the suggestion, we have acknowledged this important finding in the revised manuscript.

Page 11, line 229:

“In summary, the comparative analysis of CtNTH and AdaV with dGMP and dAMP substrates reveals two crucial points. Firstly, the binding pose of the 5'-phosphate group plays a critical role in controlling the reactivity of the nucleotide substrate. This finding aligns with a previous study that highlighted the essentiality of the 5'-phosphate moiety in dAMP for the enzymatic activity of AdaV²⁷.”

9- Figure 3a and 3b: For completeness, please include R99 and the second iron coordinating histidine, H253, in the relevant figures. Please note that H253 should be shown in all figures presented in the manuscript and supplementary information.

Response: Thank you for the suggestion. R99 and H253 have been included in the relevant figures.

But in Figure 3b, the active site is too small to show the key binding residues if we include R99 as shown below. Therefore, we did not show R99 in Figure 3b, but we have cited Figure S23, which includes R99, in the legend of Figure 3.

Page 8, line 179:

“Arg99 is too far away to be included in the active site-focused figure. The figure including Arg99 can be found in SI (Fig. S23).”

10.- It's not clear from the manuscript that the activity of AdaV towards dGMP was experimentally tested in this study. The corresponding supplementary figure should be mentioned in brackets here.

Response: Thank you for pointing out this issue. The corresponding supplementary figure (Figure S37 and S39) is now mentioned in main text. The reaction of AdaV towards dGMP is also presented in Figure 5d in the main text.

Supplementary Information:

- All HPLC chromatograms should include a scale.

Response: Thank you for your suggestions. The scale has been added to all HPLC chromatograms.

- Figure S2_ Please revise the figure legend and correct numbering.

Response: Revisions and corrections have been done.

Editorial comments:

Page 1, line 1

The authors should consider changing the title to “Discovery and Substrate Specificity Engineering of Nucleotide Halogenases”.

Page 1, line 13

“The direct C2'-halogenation of the nucleotide 2'-deoxyadenosine-5'-monophosphate (2'dAMP) has recently been achieved using the Fe(II)/ α -ketoglutarate-dependent nucleotide halogenase AdaV.”

Page 1, line 15

“However, the limited substrate scope of this enzyme hampers its broader applications.”

Page 1, line 17

“In this study, we report two novel halogenases capable of halogenating 2'-deoxyguanosine monophosphate (dGMP), thereby expanding the family of nucleotide halogenases.

Page 1, line 19

“Computational studies reveal that nucleotide specificity is regulated by the binding pose of the phosphate group. Based on these findings, we successfully engineered the substrate specificity of these halogenases by mutating second-sphere residues.”

Page 1, line 22

“This work expands the toolbox of nucleotide halogenases and provides a new strategy for nucleotide specificity engineering.”

Page 1, line 27

“Late-stage functionalization of C-H bonds has proven to be an efficient method for modifying complex molecules.”

Page 2, line 47

“The recent discovery of the Fe(II)/ α -ketoglutarate-dependent radical nucleotide halogenases, AdaV, provides a late-stage sp³ C–H halogenation method that enables direct modification of the deoxyribose ring at the C2' position with high regio- and stereoselectivity (Fig. 1c).

Page 2, line 54

“To address this limitation, new α KGHs that can act on alternative nucleotide substrates are highly sought after.”

Page 2, line 55

The authors should consider removing this sentence as it implies that a more a more comprehensive engineering campaign was carried to expand the substrate scope of these enzymes.

Page 2, line 57

“In this work, we reported the discovery and characterization of two novel Fe(II)/ α -ketoglutarate-dependent halogenases, which are capable of halogenating dGMP.”

Page 3, line 60

“Based on the nucleotide discrimination mechanism, we successful engineered switched their substrate specificity.”

Page 3, line 68

“To search for potential nucleotide halogenases candidates, we performed a sequence-based BLAST (basic local alignment search tool) analysis in NCBI (national center for biotechnology information) using AdaV as the query.”

Page 5, line 112

“Overall, through bioinformatic and biochemical analysis, we have identified two new dGMP halogenases that also display bromination and azidation activities.”

Page 7, line 149

“methylene linker”

Page 9, 194

““The divergent, productive, binding modes observed in AdaV/dAMP and CtNTH/dGMP systems can be explained by the fact that the amino groups of the adenine and guanidine moities are located at different positions on the purine ring, C6 and C2 respectively.”

Page 11, line 224

Please re-phrase this sentence. In its current form it's not coherent what the authors are trying to convey.

Page 14, line 301

“While AdaV has provided a valuable method for the direct C2'-halogenation of dAMP, its narrow substrate scope limits its applications.”

Response: We sincerely appreciate the reviewer for his/her careful reading. All modifications have been completed in accordance with the editorial comments.

Reviewer #3 (Remarks to the Author):

Ni et al report a study of an interesting family of Fe/2OG radical halogenases that utilize nucleotide substrates that was first established upon the discovery of a family member that utilizes dAMP as a substrate. They characterize the additional family members and find that dGMP is preferred for a subset, which I find to be quite interesting. They utilize MD studies to generate a hypothesis for the mode of binding of the substrate relative to the Fe center, yielding a reasonable structure that shows potential fluctuation between two nucleotide conformations. However, I do not have the expertise or training to critically judge the MD simulations. By use of sequence alignment, they identify residues in the second sphere that differ between the dAMP and dGMP halogenases, which are proposed to control the substrate selectivity. They test this hypothesis by swapping the residues in the dGMP halogenase for the corresponding amino acids found in the dAMP halogenase. They further utilize MD to explore the change in selectivity of their dGMP halogenase to favor dAMP and note that the population of the two possible nucleotide conformations has shifted.

Overall I think that this paper is interesting as it identifies new substrates for this family of radical halogenases and gives information on the nucleotide binding selectivity, which could be valuable for biocatalytic applications of these enzymes.

Response: Thank you for the positive comments.

However, the claims are too strongly worded for me throughout the engineering and mechanistic sections. I believe that these data can be described more thoughtfully without altering the impact of the paper but I am uncomfortable with some of the conclusions as written.

Response: We sincerely appreciate your professional comments. They were very useful and helpful for us for improving our manuscript. Following these valuable comments, we have carefully rephrased the conclusions. Thank you once again for your guidance and input.

1. The claim is made that a halogenase with swapped selectivity has been engineered. I apologize if I have missed some data in the SI, however the data in Figure 5 shows kinetics for the single mutant and then endpoint analysis for the double mutant. The kinetic data shows that the H274Y mutant shows a 20 fold decrease in k_{cat} for dGMP and a 2 fold increase in k_{cat} for dAMP. However, the k_{cat} for dAMP remains almost an order of magnitude below that for the wild-type enzyme for dGMP, so I do not believe one can say that the selectivity is swapped. From this data, I feel that one can only conclude that H274 appears to participate in substrate selectivity and when it is mutated to Y that both activity and selectivity is lost. Second, there is no full steady-state analysis of either H274Y or the double mutant. For the double mutant, I only see endpoint analysis. Both of these mutants need to have full steady-state analysis provided. Currently, it is impossible to assess the success of the swap and whether it is mostly derived from a catalytic defect that more strongly affects dGMP compared to dAMP because dGMP is the native substrate (a common finding).

Response: (1) We appreciate your insightful comments. In response to your suggestions,

we have conducted a full steady-state analysis of CtNTH^{H274Y} towards both substrates, which is included in the table below. Upon reevaluation of the data, we agree with your assessment that describing the selectivity as swapped is not appropriate, as the catalytic efficiency of CtNTH^{H274Y} towards dAMP remains significantly lower than that of the wild-type enzyme for dGMP. Therefore, we have revised our conclusions to reflect this more accurately.

	1 (dGMP)			3 (dAMP)		
	k_{cat} (min ⁻¹)	K_m (mM)	k_{cat}/K_m (mM ⁻¹ min ⁻¹)	k_{cat} (min ⁻¹)	K_m (mM)	k_{cat}/K_m (mM ⁻¹ min ⁻¹)
CtNTH ^{WT}	4.10 ± 0.2	0.23 ± 0.04	17.8 ± 0.3	0.55 ± 0.07	0.63 ± 0.1	0.87 ± 0.09
CtNTH ^{H274Y}	0.86 ± 0.09	1.83 ± 0.2	0.47 ± 0.08	0.83 ± 0.1	0.48 ± 0.06	1.83 ± 0.06

Page 13, line 275:

“Intriguingly, H274Y alters the substrate preference of CtNTH, showing increased activity towards dAMP compared to dGMP (Fig. 5c). This suggests that His274 plays a role in substrate selectivity. The full steady-state analysis of CtNTH^{H274Y} unveils that the altered substrate preference is primarily due to the substantial decrease in activity towards dGMP. Compared to wild-type CtNTH, the k_{cat} decreases by 5-fold, and K_m increases by 8-fold, resulting in a substantial decrease (38-fold) in catalytic efficiency (k_{cat}/K_m) towards dGMP for the H274Y variant. By comparison, CtNTH^{H274Y} exhibits only a 2-fold increase in catalytic efficiency towards dAMP.”

(2) We apologize that we made a mistake with AdaV^{A272V_Y273H} variant in the previous manuscript. It did not change the nucleotide specificity. With further mutagenesis experiments, we have confirmed that the F303V variant alters the substrate preference of AdaV. The kinetic parameters for both substrates are as follows:

	1 (dGMP)			3 (dAMP)		
	k_{cat} (min ⁻¹)	K_m (mM)	k_{cat}/K_m (mM ⁻¹ min ⁻¹)	k_{cat} (min ⁻¹)	K_m (mM)	k_{cat}/K_m (mM ⁻¹ min ⁻¹)
AdaV ^{WT}	n.d.	n.d.	n.d.	0.69 ± 0.03	1.56 ± 0.2	0.44 ± 0.1
AdaV ^{F303V}	0.79 ± 0.05	0.20 ± 0.02	4.0 ± 0.1	0.54 ± 0.02	0.24 ± 0.02	2.30 ± 0.05

Differs from CtNTH^{H274Y}, F303V variant increases the activity of AdaV towards both substrates. The shift in substrate preference for AdaV is primarily attributed to the more pronounced enhancement in activity towards the unnatural substrate, dGMP.

Furthermore, computational analysis demonstrates that the mutation of the C-terminal residue F303 alters the binding mode distribution, aligning with our previous conclusions. Thus, the relevant corrections do not affect the overall conclusion of this work. Additionally, the essential role of C-terminal residues in regulating the substrate selectivity of other halogenases (i.e., AmbO5/WelO5, HalB/HalD) has been reported by Professor Michelle C. Y. Chang, Professor Xinyu Liu and their co-workers.

2. The claim is made that: “Altogether, these results conclude that the second-sphere residues confer the nucleotide specificity by regulating the binding mode, which in turn validates our engineering strategy.” This claim is also very strongly worded when there is very little experimental support for this claim given that it is based on simulation as well as incompletely characterized mutants that do not necessarily clearly show a mechanistically valid swap in selectivity. I think this can be reworded easily to say something more along the lines that these results suggest that nucleotide specificity is regulated by the binding mode. However, I am not sure if it also validates the engineering strategy.

Response: We agree with the reviewer. The statement has been modified accordingly.

Page 16, line 358:

“Altogether, these results conclude that the second-sphere residues confer the nucleotide specificity by regulating the binding mode.”

REVIEWERS' COMMENTS

Reviewer #1 (Remarks to the Author):

The authors make appropriate and comprehensive revision and responded thoughtfully to my comments. I would recommend publication.

As a comment for the authors in their future work: in order three different simulation to be combined in one trajectory (as they currently did with 500ns runs) they need to run the simulations consequently, starting with the last saved velocities from the previous simulation.

Reviewer #2 (Remarks to the Author):

In the current version of the manuscript, Ni et al have carefully addressed my concerns regarding the readability of the manuscript and have also included the additional data that I requested in my initial review. The quality of the manuscript has been greatly improved and I support publication of the manuscript in Nature Communications. I have made some final minor editorial comments below, which may help to improve the manuscript further.

Please cite which tool was used to model CtNTH.

Page 13, line 288

Please consider re-phrasing to the following: "It was found that AdaV is highly sensitive to mutation as the designed mutants (N175I, A272V, Y273H, I299V, R302G, and F303L) significantly diminish halogenation activity towards dAMP (Fig. S37)."

Page 14, line 300

Please consider removing this sentence. "The change in substrate preference can be attributed to the significant increase in activity towards dGMP". You have already stated on Page 13, line 295 that: "To our satisfaction, the F303V variant significantly increases the activity towards dGMP and alters the substrate preference, as depicted in Fig. 5c."

Page 11, line 229:

Please consider re-phrasing to: "This finding aligns with a previous study that highlighted that the 5'-phosphate moiety in dAMP is essential for enzymatic activity of AdaV."

REVIEWERS' COMMENTS

Reviewer #1 (Remarks to the Author):

The authors make appropriate and comprehensive revision and responded thoughtfully to my comments. I would recommend publication.

As a comment for the authors in their future work: in order three different simulation to be combined in one trajectory (as they currently did with 500ns runs) they need to run the simulations consequently, starting with the last saved velocities from the previous simulation.

Response: We appreciate the positive response of our work and useful suggestions. We will take your feedback into consideration for our future work.

Reviewer #2 (Remarks to the Author):

In the current version of the manuscript, Ni et al have carefully addressed my concerns regarding the readability of the manuscript and have also included the additional data that I requested in my initial review. The quality of the manuscript has been greatly improved and I support publication of the manuscript in Nature Communications. I have made some final minor editorial comments below, which may help to improve the manuscript further.

Response: Thank you for your positive assessment of the manuscript.

Please cite which tool was used to model CtNTH.

Response: The information has been added.

Page 21, line 495:

“The structural model generated by Swiss Model was utilized for the following computational studies.”

Page 13, line 288

Please consider re-phrasing to the following: “It was found that AdaV is highly sensitive to mutation as the designed mutants (N175I, A272V, Y273H, I299V, R302G, and F303L) significantly diminish halogenation activity towards dAMP (Fig. S37).”

Response: Thank you for your careful reading. The sentence has been rephrased.

Page 14, line 300

Please consider removing this sentence. “The change in substrate preference can be attributed to the significant increase in activity towards dGMP”. You have already stated on Page 13, line 295 that: “To our satisfaction, the F303V variant significantly increases the activity towards dGMP and alters the substrate preference, as depicted in Fig. 5c.”

Response: Thank you for your suggestion. The sentence has been removed.

Page 11, line 229:

Please consider re-phrasing to: “This finding aligns with a previous study that highlighted that the 5’-phosphate moiety in dAMP is essential for enzymatic activity of AdaV.”

Response: The statement has been rephrased.